# CDC20B is required for deuterosome-mediated centriole production in multiciliated cells

Diego R. Revinski [1], Laure-Emmanuelle Zaragosi [2], Camille Boutin[1], Sandra Ruiz-Garcia[2], Marie Deprez[2], Virginie Thomé[1], Olivier Rosnet[1], Anne-Sophie Gay[2], Olivier Mercey[2], Agnès Paquet[2], Nicolas Pons[2], Gilles Ponzio[2], Brice Marcet[2], Laurent Kodjabachian [1] & Pascal Barbry [2]

Multiciliated cells (MCCs) harbor dozens to hundreds of motile cilia, which generate hydrodynamic forces important in animal physiology. In vertebrates, MCC differentiation involves massive centriole production by poorly characterized structures called deuterosomes. Here, single-cell RNA sequencing reveals that human deuterosome stage MCCs are characterized by the expression of many cell cycle-related genes. We further investigated the uncharacterized vertebrate-specific *cell division cycle 20B* (*CDC20B*) gene, which hosts microRNA-449abc. We show that CDC20B protein associates to deuterosomes and is required for centriole release and subsequent cilia production in mouse and *Xenopus* MCCs. CDC20B interacts with PLK1, a kinase known to coordinate centriole disengagement with the protease Separase in mitotic cells. Strikingly, over-expression of Separase rescues centriole disengagement and cilia production in CDC20B-deficient MCCs. This work reveals the shaping of deuterosome-mediated centriole production in vertebrate MCCs, by adaptation of canonical and recently evolved cell cycle-related molecules.

[1] Aix Marseille Univ, CNRS, IBDM, Marseille, France. [2] Université Côte d'Azur, CNRS, IPMC, Sophia-Antipolis, France. These authors contributed equally: Diego R. Revinski, Laure-Emmanuelle Zaragosi, Camille Boutin. Correspondence and requests for materials should be addressed to B.M. (email: marcet@ipmc.cnrs.fr) or to L.K. (email: laurent.kodjabachian@univ-amu.fr) or to P.B. (email: barbry@ipmc.cnrs.fr)

Multiciliated cells (MCCs) are present throughout metazoan evolution and serve functions ranging from locomotion of marine larvae and flatworms, to brain homeostasis, mucociliary clearance of pathogens and transportation of oocytes in vertebrates[1–3]. The formation of MCCs requires the production of numerous motile cilia through a complex process called multiciliogenesis[2,3]. The transcriptional control of multiciliogenesis has been decrypted to a large extent, through studies in *Xenopus* and mouse[2]. Seating at the top of the cascade, the Geminin-related factors GemC1[4–7] and Multicilin[8,9] (MCIDAS in mammals) are both necessary and sufficient to initiate MCC differentiation. GemC1 and Multicilin in complex with E2F transcription factors have been reported to activate the expression of Myb, FoxJ1, Rfx2, and Rfx3, which collectively regulate the expression of a large body of effectors required for the formation of multiple motile cilia[4,5,8–11]. Recently, defective multiciliogenesis caused by mutations in MCIDAS and Cyclin O (CCNO) has been associated with congenital respiratory and fertility syndromes in human[12,13].

Each cilium sits atop a modified centriole, called a basal body (BB). After they exit from the cell cycle, maturing MCCs face the challenge of producing dozens to hundreds of centrioles in a limited time window. In vertebrate MCCs, bulk centriole biogenesis is mostly achieved through an acentriolar structure named the deuterosome, although canonical amplification from parental centrioles also occurs[1–3]. The deuterosome was first described in early electron microscopy studies of various multiciliated tissues including the mammalian lung[14] and oviduct[15,16], the avian trachea[17], and the *Xenopus* tadpole epidermis and trachea[18]. In mammalian MCCs, the deuterosome was described as a spherical mass of fibers organized into an inner dense region and an outer, more delicate, corona[16]. In *Xenopus*, deuterosomes were initially named procentriole organizers and were reported as dense amorphous masses[18]. Recent studies have revealed that deuterosome-mediated centriole synthesis mobilizes key components of the centriole-dependent duplication pathway of the cell cycle, including CEP152, PLK4, and SAS6[19–21]. However, the deuterosome itself differs from the centriole and may contain specific components. The identification of one such component, called DEUP1 for Deuterosome assembly protein 1, opened the possibility to investigate the deuterosome at the molecular level[21]. In mouse tracheal ependymal cells, DEUP1 was detected in the core of the deuterosome[21]. DEUP1, also known as CCDC67, is a conserved vertebrate paralogue of CEP63, itself known for its importance in initiation of centriole duplication during the cell cycle[21,22]. Consistently, DEUP1 was shown to be essential for centriole multiplication in mouse and *Xenopus* MCCs[21]. Both CEP63 and DEUP1 interact with CEP152, an essential event for centriole duplication and multiplication in cycling cells and MCCs, respectively[21,22]. Once centriole multiplication is over, neosynthesized centrioles must disengage from deuterosomes and parental centrioles, convert into BBs and migrate apically to dock at the plasma membrane to initiate cilium elongation.

In this study, we aimed at better understanding deuterosome biology. We found that the gene *CDC20B* was specifically expressed in maturing MCCs during the phase of centriole multiplication. We established the corresponding CDC20B protein as an essential regulator of centriole-deuterosome disengagement. This work illustrates well the strong functional relationships that exist between centriole release from deuterosomes and centriole disengagement in mitotic cells. It also posits CDC20B as a component of a "multiciliary locus" that contains several gene products, either proteins, such as MCIDAS, CCNO or CDC20B itself, or microRNAs, such as miR-449abc, which are all actively involved into vertebrate multiciliogenesis.

## Results

**MCC single-cell transcriptome at deuterosome stage.** To identify regulators of centriole multiplication, we analyzed the transcriptome of human airway epithelial cells (HAECs) at the differentiation stage corresponding to active centriole multiplication[23] at the single-cell level (Fig. 1a). Gene expression data from 1663 cells were projected on a 2D space by *t*-distributed Stochastic Neighbor Embedding (tSNE) (Fig. 1b). We identified a small group of 37 cells corresponding to maturing MCCs engaged in deuterosome-mediated centriole amplification, as revealed by the specific expression of *MCIDAS*[8], *MYB*[24], and *DEUP1*[21] (Fig. 1c, d and Supplementary Figure 1). This subpopulation was characterized by the expression of known effectors of centriole synthesis, such as *PLK4*, *STIL*, *CEP152*, *SASS6*, but also of cell cycle regulators, such as *CDK1*, *CCNB1*, *CDC20*, *SGOL2,* and *NEK2* (Fig. 1d, Supplementary Figure 1 and Supplementary Table 1). We reasoned that uncharacterized cell cycle-related genes that are specific to this subpopulation could encode components of the deuterosome-dependent centriole amplification pathway. A particularly interesting candidate in this category was *CDC20B* (Fig. 1d), which is related to the cell cycle regulators *CDC20* and *FZR1*[25] (Supplementary Figure 2a). First, the *CDC20B* gene is present in the vertebrate genomic locus that also contains the key MCC regulators *MCIDAS*[8] and *CCNO*[13]. Coexpression of *CDC20B*, *MCIDAS*, and *CCNO* throughout HAEC differentiation was indeed observed in an independent RNA sequencing study, performed on a bulk population of HAECs (Supplementary Figure 2b). These results fit well with the observation that the promoter of human *CDC20B* was strongly activated by the MCIDAS partners E2F1 and E2F4 (Supplementary Figure 2c), as also shown in *Xenopus* by others[9] (Supplementary Figure 2d). Second, the *CDC20B* gene bears in its second intron the miR-449 microRNAs, which were shown to contribute to MCC differentiation[23,26–30]. Finally, in *Xenopus* epidermal MCCs, *cdc20b* transcripts were specifically detected during the phase of centriole amplification (Supplementary Figure 2e–m). This first set of data pointed out the specific and conserved expression pattern of *CDC20B* in immature MCCs. In the rest of this study, we analyzed the putative role of CDC20B in deuterosome-mediated centriole multiplication.

**Composition and organization of vertebrate deuterosomes.** We first conducted a series of immunofluorescence analyses to gain a better understanding of deuterosome organization in mouse ependymal and *Xenopus* epidermal MCCs as models. In wholemounts of mouse ependymal walls, mature deuterosomes revealed by DEUP1 staining appeared as circular structures around a lumen (Fig. 2a). We noticed that DEUP1 also stained fibers emanating from the core into the corona. Nascent centrioles revealed by the marker FOP were organized around the DEUP1-positive core ring. STED super-resolution microscopy helped to better appreciate the regular organization of individual FOP-positive procentrioles (Fig. 2b). Proximity labeling assays have revealed that when ectopically expressed in centrosomes CCDC67/DEUP1 is found close to Pericentrin (PCNT) and γ-tubulin, two main components of the pericentriolar material (PCM)[31]. Interestingly, we found that PCNT was present in the deuterosome corona (Fig. 2a), and STED microscopy further revealed that PCNT formed fibers around growing procentrioles (Fig. 2b). γ-tubulin staining was detected in the DEUP1-positive deuterosome core, as well as in the corona (Fig. 2a). STED microscopy indicated that PCNT and γ-tubulin stained distinct interwoven fibers in the deuterosome corona. Next, we stained immature *Xenopus* epidermal MCCs with γ-Tubulin and Centrin to reveal centriole amplification platforms. These platforms

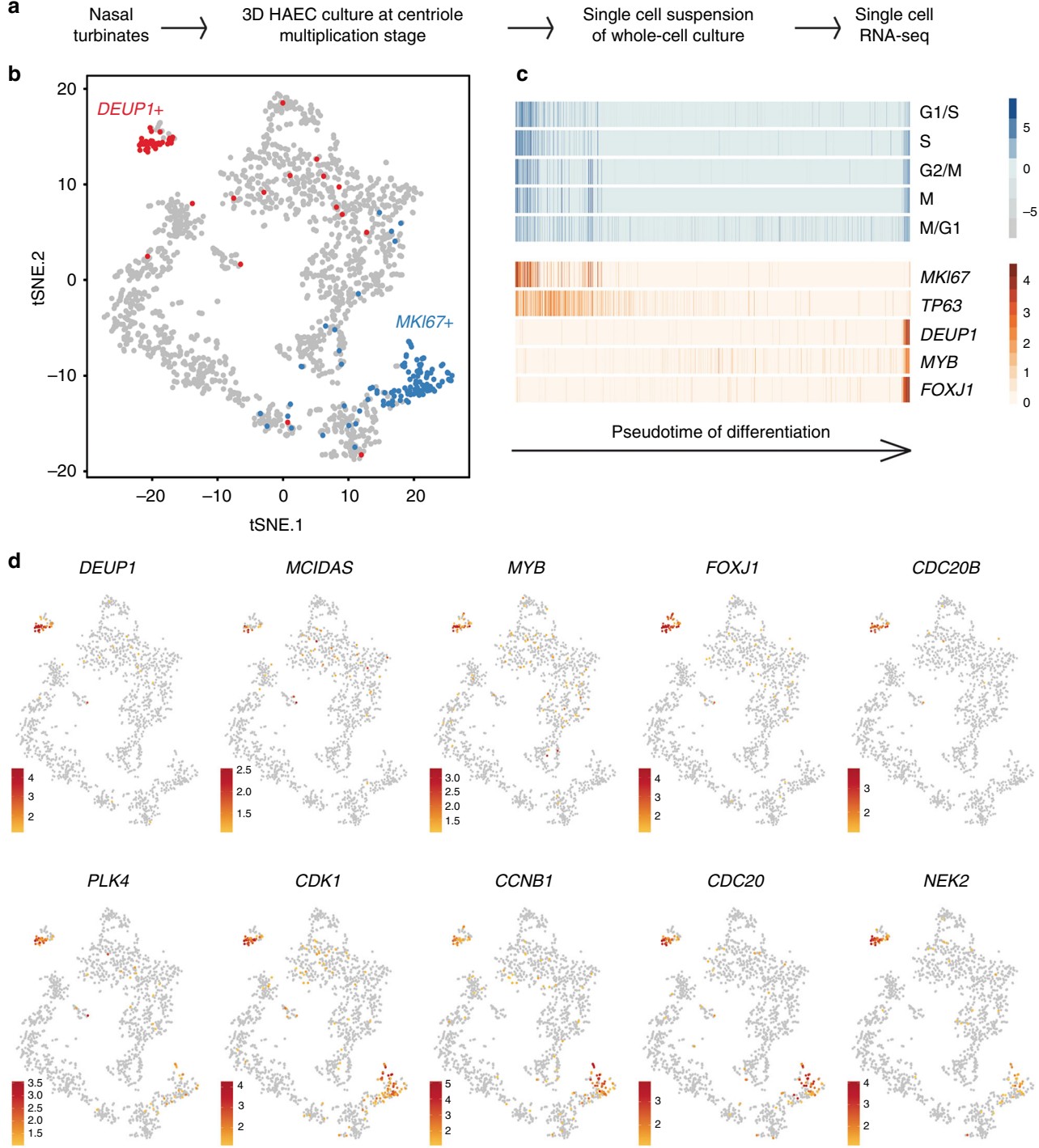

**Fig. 1** Single-cell RNA-seq analysis reveals MCC transcriptome at deuterosome stage. **a** Experimental design of the scRNA-seq experiment. **b** tSNE plot. Each point is a projection of a unique cell on a 2D space generated by the tSNE algorithm. Blue dots represent *MKI67*-positive proliferating cells, and red dots represent *DEUP1*-positive cells corresponding to maturing MCCs at deuterosome stage. **c** Cell cycle-related gene set expression in HAECs measured by scRNA-seq. Cells were ordered along a pseudotime axis, defined with the Monocle2 package. Phase-specific scores are displayed in the top heatmap. Expression of selected genes is displayed in the bottom heatmap. **d** tSNEs plots for a selection of genes specifically enriched in deuterosome stage cells. Note that *CDC20B* exhibits the most specific expression among deuterosome marker genes

displayed irregular shapes and sizes (Fig. 2c), in agreement with early electron microscopy studies[18]. Expression of low amounts of GFP-Deup1 in MCCs induced by Multicilin confirmed that active deuterosomes are embedded in γ-Tubulin-positive masses (Fig. 2d). Overall, this analysis is consistent with early ultra-structural studies, as the deuterosome core and corona can be distinguished by the presence of DEUP1 and PCNT, respectively. Moreover, γ-tubulin is a conserved marker of centriole amplification platforms in vertebrate MCCs. By analogy to the organization of the centrosome, we propose to coin the term perideuterosomal material (PDM) to describe the corona, as this region may prove important for deuterosome function.

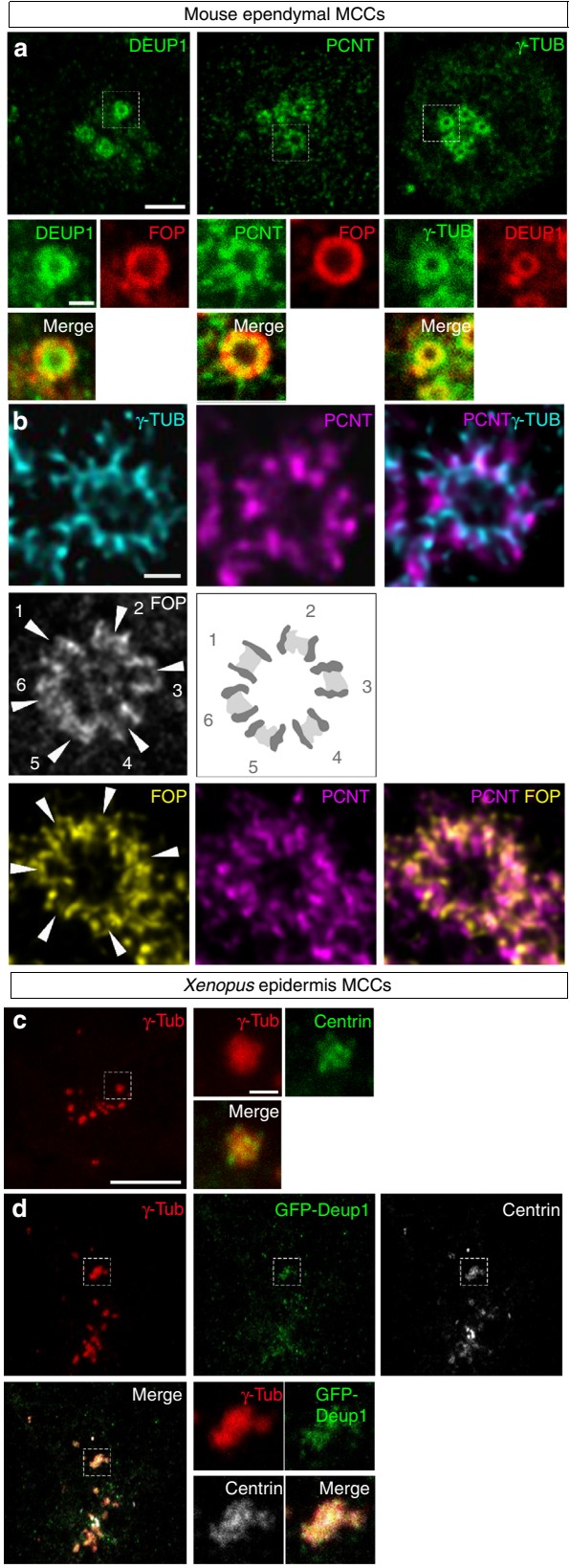

**Fig. 2** Composition and organization of vertebrate deuterosomes **a**, **b** Maturing mouse ependymal MCCs were immunostained as indicated, pictures were taken with confocal (**a**) or STED (**b**) microscope. **a** Individual deuterosomes (dashed boxes in top panels) are shown at higher magnification in bottom panels. DEUP1 stains the deuterosome core (ring) and a close fibrous area that defines the perideuterosomal region. The centriolar marker FOP reveals procentrioles arranged in a circle around the deuterosome. Pericentrin (PCNT) is enriched in the perideuterosomal region. γ-Tubulin (γ-TUB) stains the core as well as the periphery of the deuterosome. **b** STED pictures showing the organization of FOP, PCNT, and γ-TUB around deuterosomes. Individual centrioles identified by FOP staining are pointed out with arrowheads. The diagram was drawn from the adjacent FOP photograph to help reveal the regular concentric organization of nascent centrioles in a typical deuterosomal figure. **c** *Xenopus* embryos were immunostained for γ-Tubulin (γ-Tub) and Centrin and high magnification pictures of immature epidermal MCCs were taken. In these cells, Centrin-positive procentrioles grow around γ-Tubulin-positive structures. **d** *Xenopus* embryos were injected with *Multicilin-hGR* and *GFP-Deup1* mRNAs, treated with dexamethasone at gastrula st11 to induce Multicilin activity, and immunostained at neurula st18 for γ-Tubulin, GFP, and Centrin. In **c** and **d**, zooms (right panels) were made on regions identified by dashed boxes. Scale bars: 5 µm (**a**, top), 500 nm (**a**, bottom), 500 nm (**b**), 10 µm (**c**, **d**, large view), 1 µm (**c**, **d**, high magnification)

We noticed that CDC20B tended to associate primarily to large DEUP1 foci. As deuterosomes grow as they mature[21], this suggests that CDC20B may penetrate into the deuterosomal environment at a late stage of the centriole multiplication process. The same observation was made when comparing CDC20B staining in the region of immature and mature deuterosomes of mouse ependymal MCCs (Fig. 3b). As double DEUP1/CDC20B staining could not be performed on these cells, we analyzed CDC20B distribution relative to FOP-positive procentrioles. In early deuterosome stage MCCs, CDC20B was expressed at low levels and FOP staining was mostly concentrated in a large amorphous cloud (Fig. 3b). In such cells, no CDC20B staining was detected in association to FOP-positive procentrioles growing around deuterosomes. In contrast, in mature deuterosome stage MCCs, CDC20B was enriched in the innermost part of the PDM, probably very close to the deuterosome core (Fig. 3b). Further evidence was provided with a custom-made polyclonal antibody (Supplementary Figure 3b, c) used to analyze Cdc20b protein distribution in *Xenopus* epidermal MCCs. Here also, Cdc20b was found associated to Deup1-positive deuterosomes actively engaged in centriole synthesis (Fig. 3c). We finally analyzed the distribution of CDC20B in mature MCCs. As previously reported, the CDC20B protein was detected near BBs[23], but also in cilia of fully differentiated human airway MCCs (Supplementary Figure 4a–c). This was confirmed by proximity ligation assays that revealed a tight association of CDC20B with Centrin2 and acetylated α-Tubulin, in BBs and cilia, respectively (Supplementary Figure 4d–f). Fluorescent immunostaining also revealed the presence of Cdc20b in the vicinity of BBs in *Xenopus* epidermal MCCs (Supplementary Figure 4g–i). In contrast, no cilia staining was observed in these cells. Altogether, our analyses revealed that in three distinct types of MCCs in two distant vertebrate species, CDC20B is tightly associated to mature deuterosomes. We next investigated whether it may control their function.

**CDC20B associates to vertebrate deuterosomes**. We then analyzed the subcellular localization of CDC20B protein in deuterosome stage mouse and *Xenopus* MCCs. In immature mouse tracheal MCCs, double immunofluorescence revealed the association of CDC20B to DEUP1-positive deuterosomes (Fig. 3a).

**CDC20B is required for multiciliogenesis in vertebrates**. For that purpose, *Cdc20b* was knocked down in mouse ependymal MCCs, through post-natal brain electroporation of three distinct shRNAs. One of them, sh274, which targets the junction between exons 3 and 4, and can therefore only interact with mature

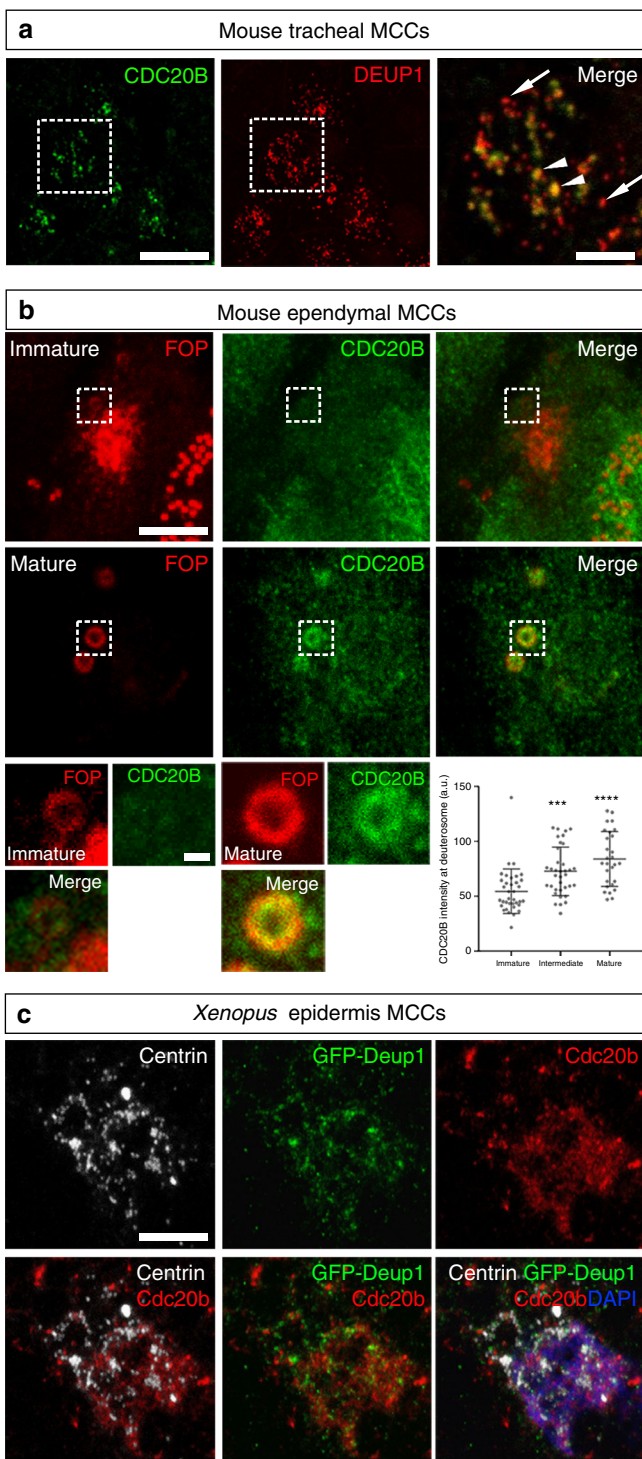

**Fig. 3** CDC20B associates to vertebrate deuterosomes. **a** Double immunofluorescence was performed on mouse tracheal MCCs after 3 days of culture in air–liquid interface. Low magnification confocal panels show coincident CDC20B and DEUP1 staining in several individual MCCs. High magnification on a single MCC reveals the prominent association of CDC20B to large deuterosomes marked by DEUP1 (arrowheads). Note that some smaller deuterosomes do not contain CDC20B (arrows). **b** Mouse ependymal MCCs were immunostained as indicated, and high magnification confocal pictures of cells with immature and mature deuterosomal figures were taken. In these cells, centrioles revealed by FOP form a ring around deuterosomes. CDC20B staining forms a ring inside the ring of FOP-positive procentrioles indicating that CDC20B is tightly associated to deuterosomes. Note that the CDC20B signal associated to deuterosome increased with their maturation (high magnification pictures of >25 cells per category from two different animals were quantified in the graph; mean values and standard deviations are shown). Unpaired *t* test vs immature: *p* = 0.0005 (intermediate, ***); *p* < 0.0001 (mature, ****). In **a** and **b**, zooms were made on regions identified by dashed boxes. **c** *Xenopus* embryos were injected with *GFP-Deup1* mRNA and immunostained at neurula st18 as indicated. Scale bars: 5 µm (**a**, **b**, large view), 1.5 µm (**a**, high magnification), 500 nm (**b**, high magnification), 10 µm (**c**)

that displayed centrioles still engaged on deuterosomes (Fig. 4f, g). Fifteen days after electroporation, a majority of CDC20B-deficient MCCs still showed a severely reduced number of released centrioles, and consequently lacked cilia (Fig. 4h–k).

*Cdc20b* was also knocked down in *Xenopus* epidermal MCCs, through injection of two independent morpholino antisense oligonucleotides targeting either the ATG (Mo ATG), or the exon 1/intron 1 junction (Mo Spl) (Supplementary Figure 5b). The efficiency of Mo ATG was verified through fluorescence extinction of co-injected Cdc20b-Venus (Supplementary Figure 5c). RT-PCR confirmed that Mo Spl caused intron 1 retention (Supplementary Figure 5d), which was expected to introduce a premature stop codon, and to produce a Cdc20b protein lacking 96% of its amino acids, likely to undergo unfolded protein response-mediated degradation. Thus, both morpholinos were expected to generate severe loss of Cdc20b function. Consistent with this interpretation, both morpholinos strongly reduced Cdc20b immunostaining in deuterosome stage MCCs (Supplementary Figure 5e). We verified that neither morpholinos caused *p53* transcript up-regulation (Supplementary Figure 5f), a non-specific response to morpholinos that is sometimes detected in zebrafish embryos[32]. Importantly, whole-mount in situ hybridization indicated that miR-449 expression was not perturbed in the presence of either morpholino (Supplementary Figure 5g). We found that *cdc20b* knockdown did not interfere with acquisition of the MCC fate (Supplementary Figure 6a–e), but severely impaired multiciliogenesis, as revealed by immunofluorescence and electron microscopy (Fig. 5a–i). This defect stemmed from a marked reduction in the number of centrioles, and poor docking at the plasma membrane (Fig. 5g–o and Supplementary Figure 6f–k). Importantly, centrioles and cilia were rescued in Mo Spl MCCs by co-injection of *cdc20b*, *venus-cdc20b* or *cdc20b-venus* mRNAs (Fig. 5j–o and Supplementary Figure 6f–k). In normal condition, *Xenopus* epidermal MCCs arise in the inner mesenchymal layer and intercalate into the outer epithelial layer, while the process of centriole amplification is underway[33]. To rule out secondary defects due to poor radial intercalation, we assessed the consequences of *cdc20b* knockdown in MCCs induced in the outer layer by Multicilin overexpression[8]. Like in natural MCCs, Cdc20b proved to be essential for the production of centrioles and cilia in response to Multicilin activity (Supplementary Figure 7a–g). We also noted that the apical actin network that

mRNA, was useful to rule out possible interference with the production of miR-449 molecules from the *Cdc20b* pre-mRNA (Supplementary Figure 5a). Five days after electroporation, all three shRNAs significantly reduced the expression of CDC20B in deuterosome stage MCCs (Fig. 4c), but did not alter MCC identity as revealed by FOXJ1 expression (Fig. 4a, b, d). Centriole production by deuterosomes was analyzed by FOP/DEUP1 double staining 9 days after electroporation. At this stage, control MCCs had nearly all released their centrioles and disassembled their deuterosomes (Fig. 4e, g). In sharp contrast, *Cdc20b* shRNAs caused a significant increase in the number of defective MCCs

normally surrounds BBs was disrupted in absence of Cdc20b, although this defect could be secondary to the absence of centrioles (Supplementary Figure 7d–g). Centrioles in Cdc20b morphant cells often formed clusters, suggesting that disengagement from deuterosomes could have failed (Fig. 5l,m). To better assess this process, we injected GFP-Deup1 in Multicilin-induced MCCs and stained centrioles with Centrin. In mature control MCCs, deuterosomes were disassembled, centrioles were converted into BBs, had docked and initiated cilium growth (Fig. 5p, s). In contrast, both morpholinos caused a marked increase in the number of defective MCCs, which were devoid of cilia and displayed centrioles still engaged on deuterosomes (Fig. 5q–u). Altogether our functional assays in mouse and *Xenopus* indicate that CDC20B is required for centriole disengagement from deuterosomes and subsequent ciliogenesis in MCCs. We next investigated the molecular mechanism of action of CDC20B underlying its role in centriole release.

**Partners and effectors of CDC20B reveal its mechanism of action**. In mitotic cells, centriole disengagement is necessary to license centriole duplication in the following cell cycle[34]. This process is known to depend on the coordinated activities of the mitotic kinase PLK1 and the protease Separase[35]. One proposed mechanism involves the phosphorylation of PCNT by PLK1, which induces its cleavage by Separase, thereby allowing centriole disengagement through disassembly of the PCM[36,37]. Separase is known to be activated by the degradation of its inhibitor Securin, which is triggered by the Anaphase Promoting Complex (APC/C) upon binding to CDC20[25]. *PLK1*, *Separase* (*ESPL1*), *Securin* (*PTTG1*), *CDC20*, and *PCNT* were all found to be expressed in human deuterosome stage MCCs (Fig. 1d and Supplementary Figure 1). We have shown above that PCNT is present in the PDM and a recent study revealed the presence of CDC20 and the APC/C component APC3 in mouse ependymal MCCs at the stage of centriole disengagement[38]. Based on this large body of information, we hypothesized that centriole-deuterosome disengagement involves the coordinated activities of PLK1 and Separase, and that CDC20B would be involved in this scenario. *CDC20B* encodes a protein of about 519 amino acids largely distributed across the vertebrate phylum[23]. In its C-terminal half, CDC20B contains seven well conserved WD40 repeats, predicted to form a β-propeller, showing 49 and 37% identity to CDC20 and FZR1 repeats, respectively (Supplementary Figure 2a). However, CDC20B lacks canonical APC/C binding domains (Supplementary Figure 2a). Using mass spectrometry on immunoprecipitated protein complexes from transfected HEK cells, we could identify multiple APC/C components interacting with CDC20 but not with CDC20B (Supplementary Table 2). We conclude that CDC20B is probably incapable of activating APC/C. Interestingly, an unbiased interactome study reported association of CDC20B with PLK1[39]. Using reciprocal co-immunoprecipitation assays in HEK transfected cells, we confirmed that CDC20B and PLK1 could be found in the same complex (Fig. 6a and Supplementary Figure 8). This suggested that CDC20B could cooperate with PLK1 to trigger centriole disengagement. Consistent with this hypothesis, we found that PLK1 was enriched in the PDM of mature deuterosomes in mouse ependymal MCCs (Fig. 6b), in agreement with a recent report[38]. Another interesting partner of CDC20B identified in a second unbiased interactome study[40] was SPAG5 (Astrin), which was reported to control timely activation of Separase during the cell cycle[41,42]. Using the same strategy as above, we could detect CDC20B and SPAG5 in the same complex (Fig. 6c and Supplementary Figure 8). As SPAG5 was found associated to DEUP1 in a proximity labeling assay[31], we assessed its localization in

deuterosomes. Strikingly, SPAG5 was detectable in mature deuterosomes of mouse ependymal MCCs, with a clear enrichment in the deuterosome core (Fig. 6d). Finally, reciprocal co-immunoprecipitations revealed that CDC20B and DEUP1 were detected in the same complex when co-expressed in HEK cells (Fig. 6e and Supplementary Figure 8). Consistent with this result, we observed that RFP-Cdc20b was recruited around spherical Deup1-GFP structures positive for γ-Tubulin and Centrin in *Xenopus* epidermal MCCs (Supplementary Figure 7h–m). This series of experiments suggested that CDC20B could participate in the assembly of a protein complex in mature deuterosomes, required to coordinate the activities of PLK1 and Separase for centriole disengagement. As Separase is the last effector in this scenario, we tested whether over-expressing human Separase in *Xenopus cdc20b* morphant MCCs could rescue centriole disengagement. In support to our hypothesis, over-expression of wild-type, but not protease-dead Separase, efficiently rescued centriole disengagement and cilia formation in *cdc20b* morphant MCCs (Fig. 7a–g and Supplementary Figure 7n–s). Separase could also rescue multiciliogenesis in Multicilin-induced MCCs injected with *cdc20b* Mos (Supplementary Figure 7t–z). We conclude that CDC20B is involved in Separase-mediated release of mature centrioles from deuterosomes in vertebrate MCCs (Fig. 7h).

**Discussion**

In this study, we report the essential and conserved role of CDC20B in vertebrate multiciliogenesis. Our data suggest that the presence of CDC20B in the perideuterosomal region is necessary to allow centriole disengagement. We note, however, that our data, which are based on partial knockdowns, remain compatible with an earlier function of CDC20B in promoting deuterosome assembly and/or activity. A total genetic knockout of *Cdc20b* should help to assess this possibility in mouse tracheal and ependymal MCCs. By analogy to mitosis, we propose that CDC20B is involved in Separase-dependent proteolysis at deuterosomes, allowing the release of mature centrioles and subsequent ciliogenesis. This view is consistent with a recent report showing that centriole disengagement in murine ependymal MCCs involves the activities of PLK1, a partner of CDC20B, and APC/C, the activator of Separase[38]. The central question arising from our work then becomes: how are CDC20B and Separase activities integrated? The simple scenario of a CDC20-like function of CDC20B is very unlikely as it does not appear to bind APC/C (Supplementary Table 2). CDC20 was detected in cultured murine ependymal MCCs during the phase of centriole disengagement[38], and FZR1 genetic ablation was reported to cause reduced production of centrioles and cilia in the same cells[43]. APC/C is therefore likely activated in maturing MCCs by its classical activators, CDC20 and/or FZR1, leading to Separase activation through degradation of its inhibitor Securin. In that context, we propose that additional factors linked directly or indirectly to CDC20B may contribute to activation of Separase. It was shown that SPAG5 inhibits or activates Separase depending on its status of phosphorylation[41,42]. As the phosphorylation status of SPAG5 was shown to be controlled by PLK1[44], our data suggest that the CDC20B/PLK1/SPAG5 complex could control the timing of Separase activation locally in deuterosomes. It is therefore possible that multiple modes of activation of Separase may act in parallel to trigger the release of neo-synthesized centrioles in maturing MCCs. Alternatively, different pathways may be used in distinct species, or in distinct types of MCCs. An important question for future studies regards the identity of PLK1 and Separase substrates involved in centriole disengagement. Work on mitotic cells[36,37] and our own analysis suggest that PCNT may represent a prime target. Another potentially relevant

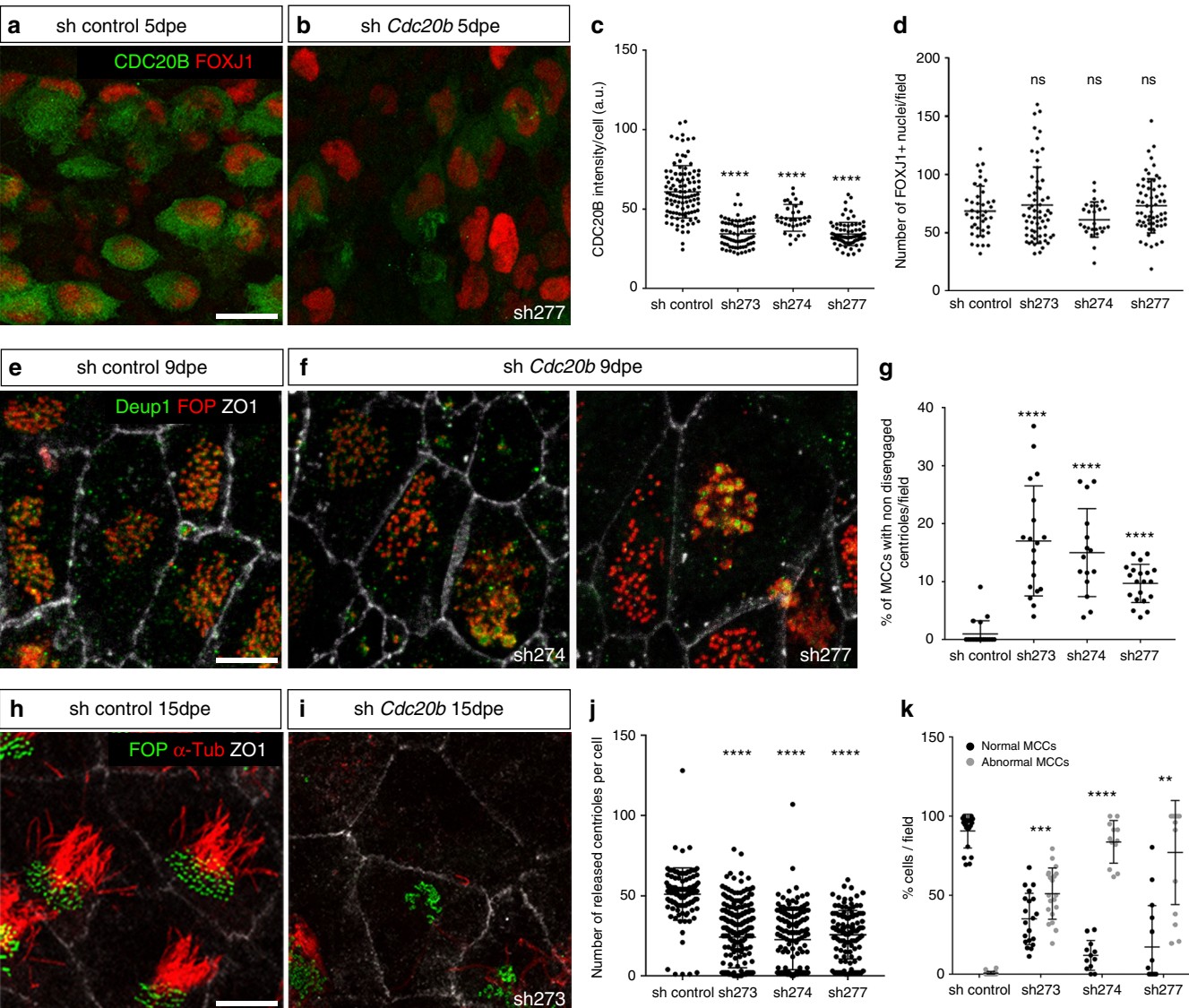

**Fig. 4** CDC20B knockdown impairs multiciliogenesis in mouse ependymal MCCs. **a**, **b** Ependyma were stained for CDC20B (green) and FOXJ1 (nuclear MCC fate marker, red) 5 days post electroporation (5dpe) of control shRNA (**a**) or *Cdc20b* shRNA (**b**). sh277 is exemplified here, but all three *Cdc20b* shRNAs produced similar effects. **c** Graph showing the quantification of CDC20B protein levels in cells at the deuterosomal stage at 5dpe from two experiments. Mean values and standard error are shown. Unpaired *t*-test: ****$p < 0.0001$. **d** Dot plot showing the number of FOXJ1-positive nuclei observed for each field, with mean values and standard deviations from two experiments. Unpaired *t*-test: $p = 0.3961$ (sh273, ns), $p = 0.1265$ (sh274, ns), $p = 0.3250$ (sh277, ns). No significant variations were observed between conditions, indicating that MCC fate acquisition was not affected by *Cdc20b* knockdown. **e**, **f** Confocal pictures of 9dpe ependyma electroporated with control shRNA (**e**) or *Cdc20b* shRNAs (**f**) and stained for DEUP1 (deuterosome, green), FOP (centrioles, red) and ZO1 (cell junction, white). DEUP1-positive deuterosomes with non-disengaged FOP-positive centrioles were observed much more frequently in MCCs electroporated with *Cdc20b* shRNAs compared to control. **g** Dot plot showing the percentage of MCCs with non-disengaged centrioles per field, with mean values and standard deviations. Two experiments were analyzed. Unpaired *t*-test: ****$p < 0.0001$. **h**, **i** Confocal pictures of 15dpe ependyma stained for FOP (centrioles, green), α-Tubulin (α-TUB, cilia, red), and ZO1 (cell junction, white) showing the morphology of normal MCCs in shRNA control condition (**h**), and examples of defects observed in MCCs treated with sh *Cdc20b* (**i**). **j** Dot plot showing the number of released centrioles per cell, with mean values and standard deviations. **k** Dot plot showing the percentage of normal and abnormal MCCs per field of observation, with mean values and standard deviations. MCCs were scored abnormal when they did not display organized centriole patches associated to cilia. Three experiments were analyzed. Unpaired *t*-test: $p = 0.0004$ (sh273, ***), $p = 0.0001$ (sh274, ****), $p = 0.0038$ (sh277, **). Scale bars: 20 μm (**a**), 5μm (**e**, **i**)

candidate could be DEUP1 itself as it is clear that deuterosomes are disassembled after the release of centrioles. In that respect, it is interesting to note the presence of multiple PLK1 consensus phosphorylation sites in human, mouse, and *Xenopus* DEUP1.

In this study, we have introduced the notion of perideuterosomal material, in analogy to the pericentriolar material. It is

striking that the two main components of the PCM, PCNT, and γ-Tubulin, are also present in the PDM, which begs the question whether additional PCM proteins may be present in the PDM. The PDM may constitute a platform to sustain procentriole growth, through the concentration and delivery of elementary parts. It could also have a mechanical role to hold in place the

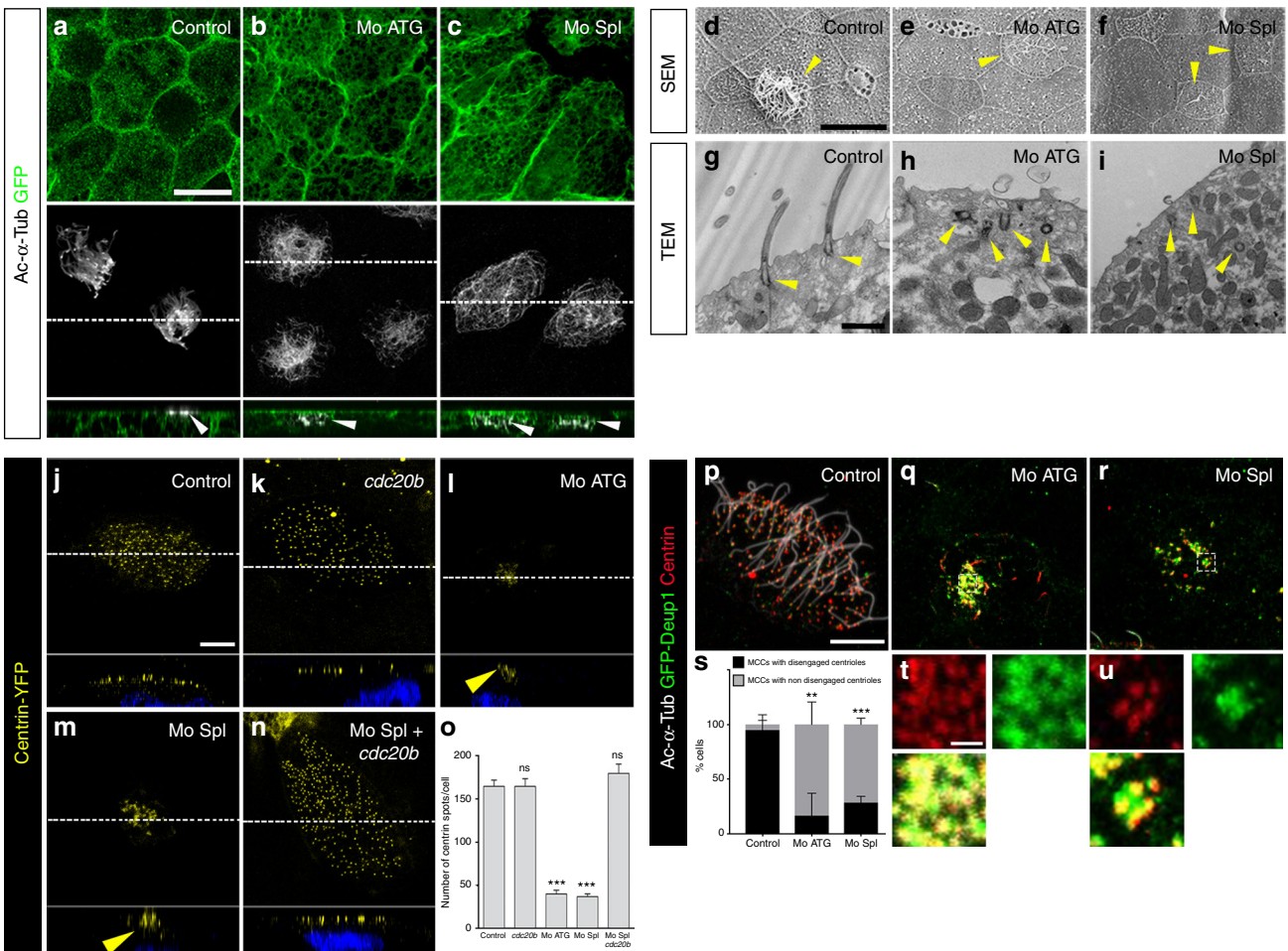

**Fig. 5** *cdc20b* knockdown impairs multiciliogenesis in *Xenopus* epidermal MCCs. **a–c** 8-cell embryos were injected in presumptive epidermis with *GFP-CAAX* mRNA and *cdc20b* morpholinos, as indicated. Embryos at tailbud st25 were processed for fluorescent staining against GFP (injection tracer, green) and Acetylated-α-Tubulin (Ac-α-Tub, cilia, white). White dotted lines indicate the position of orthogonal projections shown in bottom panels. Note that *cdc20b* morphant MCCs display cytoplasmic filaments but do not grow cilia (white arrowheads). **d–f** Scanning electron microscopy (SEM) of control (**d**) and *cdc20b* morphant (**e**, **f**) embryos at tadpole st31. Yellow arrowheads point at normal (**d**) and defective MCCs (**e**, **f**). **g–i** Transmission electron microscopy (TEM) of control (**g**) and *cdc20b* morphant (**h**, **i**) embryos at tailbud st25. Yellow arrowheads point at normally docked basal bodies supporting cilia (**g**) and undocked centrioles unable to support cilia (**h**, **i**). **j–n** 8-cell embryos were injected in presumptive epidermis with *centrin-YFP* mRNA, *cdc20b* morpholinos, and *cdc20b* mRNA, as indicated. Centrin-YFP fluorescence was observed directly to reveal centrioles (yellow). Nuclei were revealed by DAPI staining in blue. White dotted lines indicate the position of orthogonal projections shown in bottom panels. Yellow arrowheads point at undocked centrioles. **o** Bar graph showing the mean number of BBs per MCC, and standard error mean, as counted by Centrin-YFP dots. One-way ANOVA and Bonferroni's multiple comparisons test on two experiments, ***$p < 0.0001$. *cdc20b* knockdown significantly reduced the number of BBs per cell, and this defect could be corrected by *cdc20b* co-injection with Mo Spl. **p–u** Embryos were injected with *Multicilin-hGR* and *GFP-Deup1* mRNAs, treated with dexamethasone at gastrula st11 to induce Multicilin activity, and immunostained at neurula st23 against Acetylated-α-tubulin (cilia, white), GFP (deuterosomes, green), and Centrin (centrioles, red). **p** Control cells showed individual centrioles, many of which had initiated ciliogenesis. Note that Deup1-positive deuterosomes were no longer visible at this stage. (**q**, **r**, **t**, **u**) *cdc20b* morphant MCCs showed procentrioles still engaged on deuterosomes and lacked cilia. In **t** and **u**, zooms were made on regions identified by dashed boxes in **q** and **r**. **s** Bar graph showing the mean percentage of cells that completed or not centriole disengagement with standard deviations. Three experiments were analyzed. Unpaired *t*-test: $p = 0.0037$ (Mo ATG, **), $p = 0.0004$ (Mo Spl, ***). Scale bars: 20 μm (**a**, **d**), 1 μm (**g**, **t**), 5 μm (**j**, **p**)

growing procentrioles. Future work should evaluate deuterosome-mediated centriole synthesis in absence of major PDM components.

We found that beyond its association to deuterosomes during the phase of centriole amplification, CDC20B was also associated to BBs and cilia in fully differentiated mammalian MCCs. This dual localization is consistent with failed ciliogenesis upon CDC20B knockdown in mouse ependymal MCCs. However, while we could detect Cdc20b near BBs of mature MCCs in *Xenopus*, we found no evidence of its presence in cilia. Furthermore, cilia were rescued by Separase overexpression in Cdc20b

morphant MCCs. This suggests that Cdc20b is not required for ciliogenesis in this species, although it could potentially contribute to cilium structure and/or function. Thus, refined temporal and spatial control of CDC20B inhibition will be needed to study its function beyond centriole synthesis.

This and previous studies[23,26–28] establish that the miR-449 cluster and its host gene *CDC20B* are commonly involved in multiciliogenesis. Consistent with its early expression, it was suggested that miR-449 controls cell cycle exit and entry into differentiation of MCCs[23,27,30]. This study reveals that CDC20B itself is involved in the production of centrioles, the first key step

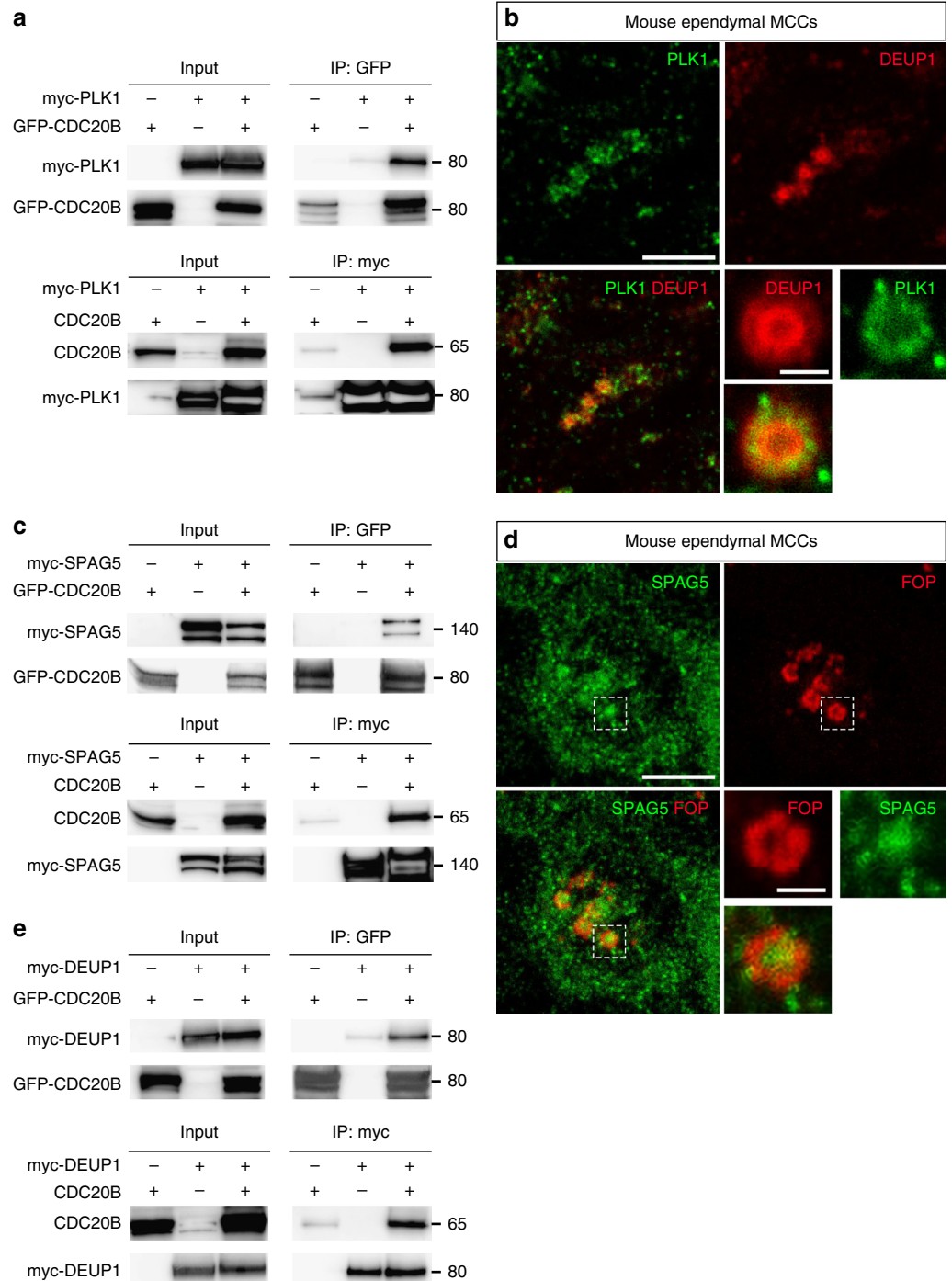

**Fig. 6** CDC20B interacts with PLK1, SPAG5, and DEUP1. **a**, **c**, **e** Co-immunoprecipitations of PLK1, SPAG5, and DEUP1 with CDC20B were tested after transfections of different constructs in HEK cells, indicated at the top of each panel. Proteins (left legend) were revealed by immunoblotting. Representative blots from 3 independent experiments. **b**, **d** Maturing mouse ependyma were immunostained for the indicated proteins, and pictures were taken with a confocal microscope. PLK1 and SPAG5 are expressed in maturing MCCs. In **b**, zoom was made on a different cell to reveal PLK1 enrichment in the perideuterosomal region. In **d**, zoom was made on the dashed box to reveal SPAG5 enrichment in the deuterosome core. Scale bars in **b**, **d**: 5 μm (large view), 1 μm (high magnification)

of the multiciliogenesis process. From that perspective, the nested organization of miR-449 and *CDC20B* in vertebrate genomes, which allows their coordinated expression, appears crucial for successful multiciliogenesis.

It is also noteworthy to point out the location of this gene in a genomic locus where congenital mutations in MCIDAS and CCNO were recently shown to cause a newly-recognized MCC-specific disease, called reduced generation of multiple motile cilia (RGMC). RGMC is characterized by severe chronic lung infections and increased risk of infertility[12,13]. Its location in the same genetic locus as MCIDAS and CCNO makes CDC20B a putative candidate for RGMC. By extension, the deuterosome stage-specific genes uncovered by scRNA-seq in this study also represent potential candidates for additional RGMC mutations.

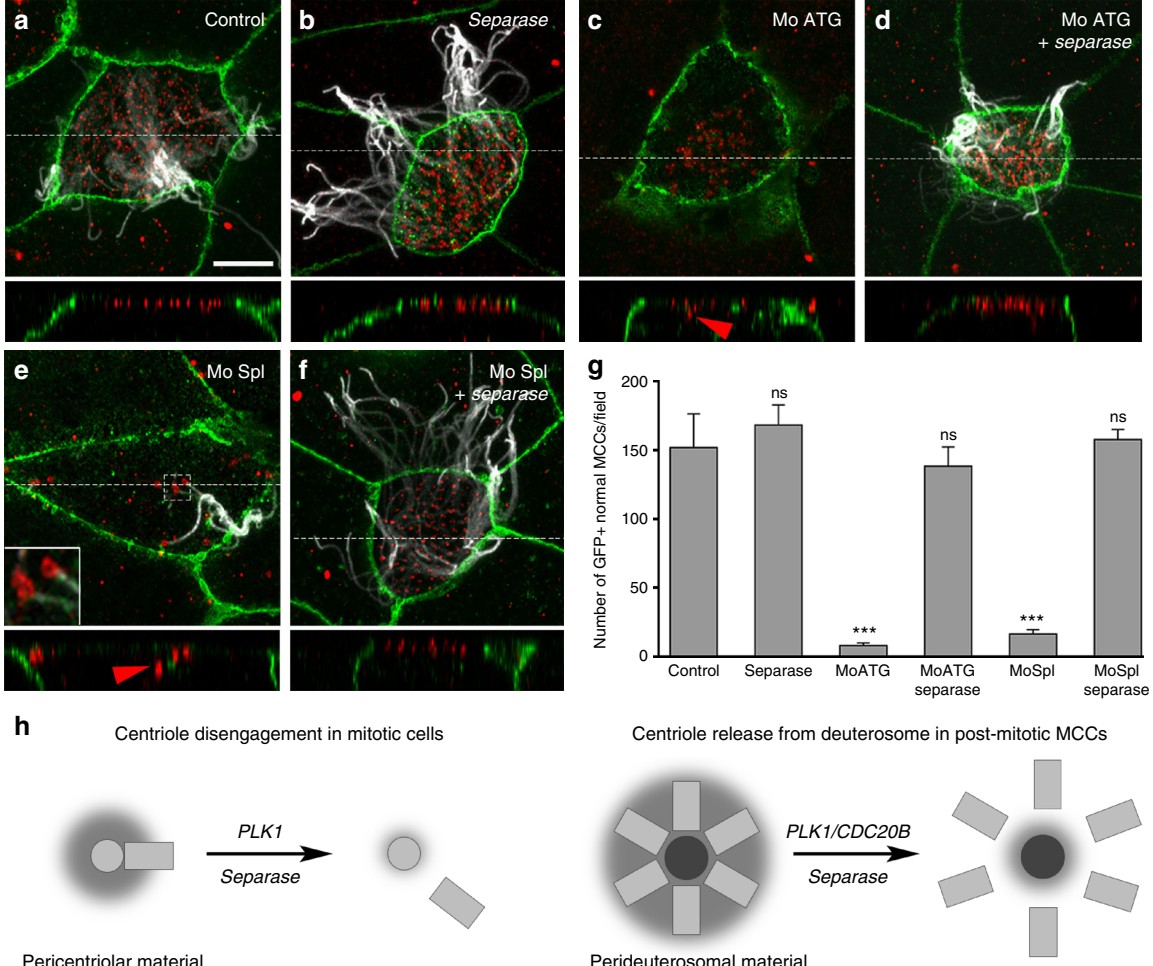

**Fig. 7** Separase overexpression rescues multiciliogenesis in absence of Cdc20b. **a–f** 8-cell *Xenopus* embryos were injected in the presumptive epidermis with *GFP-gpi* mRNA, *cdc20b* morpholinos, and human *Separase* mRNA, as indicated. Embryos were fixed at tailbud st25 and immunostained against GFP (injection tracer, green), Acetylated-α-Tubulin (cilia, white) and ɣ-Tubulin (BBs, red). White dotted lines indicate the position of orthogonal projections shown in bottom panels. Red arrowheads point undocked BBs. Left inset in **e** shows zoom on clustered centrioles. **g** Bar graph showing the mean number of properly ciliated MCCs among injected cells, per field of observation, with standard error mean, from two independent experiments. Counting was performed on pictures taken at low magnification (×20), in order to score a large number of cells. Separase overexpression fully rescued multiciliogenesis in *cdc20b* morphant MCCs. One-way ANOVA and Bonferroni's multiple comparisons on two experiments, ***$p < 0.0001$, $^{ns}p > 0.05$. Scale bars: 5 μm (**a**). **h** Model illustrating the analogy between centriole disengagement in mitotic cells and centriole release from deuterosomes in post-mitotic MCCs

Previous works have established the involvement of the centriole duplication machinery active in S-phase of the cell cycle, during centriole multiplication of vertebrate post-mitotic MCCs[19–21]. Our study further reveals a striking analogy between centriole disengagement from deuterosomes in MCCs, and centriole disengagement that occurs during the M/G1 transition of the cell cycle (Fig. 7g). Thus, it appears that centriole production in MCCs recapitulates the key steps of the centriole duplication cycle[34]. However, the cell cycle machinery must adapt to the acentriolar deuterosome to massively produce centrioles. Such adaptation appears to involve physical and functional interactions between canonical cell cycle molecules, such as CEP152 and PLK1, and recently evolved cell cycle-related deuterosomal molecules, such as DEUP1[21] and CDC20B. It remains to examine whether additional deuterosomal cell cycle-related molecules have emerged in the vertebrate phylum to sustain massive centriole production.

In conclusion, this work illustrates how coordination between ancestral and recently evolved cell cycle-related molecules can give rise to a novel differentiation mechanism in vertebrates.

## Methods

**Subjects/human samples**. Inferior turbinates were from patients who underwent surgical intervention for nasal obstruction or septoplasty (provided by L. Castillo, Nice University Hospital, France). Experiments involving human tissues were performed according to the guidelines of the Declaration of Helsinki, after approval by the institutional review board "Comité de Protection des Personnes Sud Méditerranée V" (06/16/2015). All patients gave their written informed consent.

**Single-cell RNA sequencing of human airway epithelial cells (HAECs)**. HAECs cultures were derived from nasal mucosa of inferior turbinates. After excision, nasal inferior turbinates were immediately immersed in $Ca^{2+}/Mg^{2+}$-free HBSS supplemented with 25 mM HEPES, 200 U/mL penicillin, 200 μg/mL streptomycin, 50 μg/mL gentamicin sulfate, and 2.5 μg/mL amphotericin B (all reagents from Gibco). After repeated washes with cold supplemented HBSS, tissues were digested with 0.1% Protease XIV from *Streptomyces griseus* (Sigma) overnight at 4 °C. After incubation, fetal calf serum (FCS) was added to a final concentration of 10%, and nasal epithelial cells were detached from the stroma by gentle agitation. Cell suspensions were further dissociated by trituration through a 21 G-needle and then centrifuged at 150×*g* for 5 min. The pellet was resuspended in supplemented HBSS containing 10% FCS and centrifuged again. The second cell pellet was then suspended in Dulbecco's Modified Eagle's Medium (DMEM, Gibco) containing 10% FCS and cells were plated (20 000 cells per cm²) on 75 cm²-flasks coated with rat tail collagen I (Sigma-Aldrich). Cells were incubated in a humidified atmosphere of 5% CO₂ at 37 °C. Culture medium was replaced with Bronchial Epithelium Basal Medium (BEBM, Lonza) supplemented with BEGM SingleQuot Kit Supplements

(Lonza) on the day after and was then changed every other day. After 4 to 5 days of culture, after reaching about 70% confluence, cells were detached with trypsin-EDTA 0.05% (Gibco) for 5 min and seeded on Transwell® permeable supports (6.5 mm diameter; 0.4 μm pore size; Corning), in BEGM medium, with a density of 30,000 cells per Transwell®. Once the cells have reached confluence (typically after 5 days), they were induced to differentiate at the air–liquid interface by removing medium at the apical side of the Transwell®, and by replacing medium at the basal side with DMEM:BEBM (1:1) supplemented with BEGM SingleQuot Kit Supplements. Culture medium was changed every other day. Single-cell analysis was performed after 14 days of culture at the air–liquid interface, which corresponds to the maximum centriole multiplication stage. To obtain a single-cell suspension, cells were incubated with 0.1% protease type XIV from S. griseus in supplemented HBSS for 4 h at 4 °C. Cells were gently detached from Transwells® by pipetting and then transferred to a microtube. 50 units of DNase I (EN0523 ThermoFisher Scientific) per 250 μL were directly added and cells were further incubated at room temperature for 10 min. Cells were centrifuged (150×g for 5 min) and resuspended in 500 μL supplemented HBSS containing 10% FCS, centrifuged again (150×g for 5 min) and resuspended in 500 μL HBSS before being mechanically dissociated through a 26 G syringe (4 times). Finally, cell suspensions were filtered through a Scienceware® Flowmi™ Cell Strainer (40 μm porosity), centrifuged (150×g for 5 min) and resuspended in 500 μL of cold HBSS. Cell concentration measurements were performed with Scepter™ 2.0 Cell Counter (Millipore) and Countess™ automated cell counter (ThermoFisher Scientific). Cell viability was checked with Countess™ automated cell counter (ThermoFisher Scientific). All steps except the DNAse I incubation were performed on ice. For the cell capture by the 10× genomics device, the cell concentration was adjusted to 300 cells/μL in HBSS aiming to capture 1500 cells. We then applied the manufacturer's protocol (Chromium™ Single Cell 3′ Reagent Kit, v2 Chemistry) to obtain single cell 3′ libraries for Illumina sequencing. Libraries were sequenced with a NextSeq 500/550 High Output v2 kit (75 cycles) that allows up to 91 cycles of paired-end sequencing: the forward read had a length of 26 bases that included the cell barcode and the UMI; the reverse read had a length of 57 bases that contained the cDNA insert. CellRanger Single-Cell Software Suite v1.3 was used to perform sample demultiplexing, barcode processing and single-cell 3′ gene counting using default parameters and human build hg19. Additional analyses were performed using R. Pseudotemporal ordering of single cells was performed with the last release of the Monocle package[45]. Cell cycle scores were calculated by summing the normalized intensities of genes belonging to phase-specific gene sets then centered and scaled by phase. Gene sets for each phase were curated from previously described sets of genes[46] (Table S2). Data was submitted to the GEO portal under series reference GSE103518. Data shown in Fig. 1 is representative of four independent experiments performed on distinct primary cultures.

**RNA sequencing of HAECs.** For Supplementary Fig. 2B, three independent HAEC cultures (HAEC1, HAEC2, HAEC3) were triggered to differentiate in air–liquid interface (ALI) cultures for 2 days (ALI day 2, undifferentiated), ALI day 14 (first cilia), or ALI day 28 (well ciliated). RNA was extracted with the miRNeasy mini kit (Qiagen) following manufacturer's instructions. mRNA-seq was performed from 2 μg of RNA that was first subjected to mRNA selection with Dynabeads® mRNA Purification Kit (Invitrogen). mRNA was fragmented 10 min at 95 °C in RNAseIII buffer (Invitrogen) then adapter-ligated, reverse transcribed and amplified (6 cycles) with the reagents from the NEBNext Small RNA Library Prep Set for SOLiD. Small RNA-seq was performed from 500 ng RNA with the NEBNext Small RNA Library Prep Set for SOLiD (12 PCR cycles) according to manufacturer's instructions. Both types of amplified libraries were purified on Purelink PCR micro kit (Invitrogen), then subjected to additional PCR rounds (8 cycles for RNA-seq and 4 cycles for small RNA-seq) with primers from the 5500 W Conversion Primers Kit (Life Technologies). After Agencourt® AMPure® XP beads purification (Beckman Coulter), libraries were size-selected from 150 nt to 250 nt (for RNA-seq) and 105 nt to 130 nt (for small RNA-seq) with the LabChip XT DNA 300 Assay Kit (Caliper Lifesciences), and finally quantified with the Bioanalyzer High Sensitivity DNA Kit (Agilent). Libraries were sequenced on SOLiD 5500XL (Life Technologies) with single-end 50b reads. SOLiD data were analyzed with lifescope v2.5.1, using the small RNA pipeline for miRNA libraries and whole transcriptome pipeline for RNA-seq libraries with default parameters. Annotation files used for production of raw count tables correspond to Refseq Gene model v20130707 for mRNAs and miRBase v18 for small RNAs. Data generated from RNA sequencing were then analyzed with Bioconductor (http://www.bioconductor.org) package DESeq and size-factor normalization was applied to the count tables. Heatmaps were generated with GenePattern using the "Hierarchical Clustering" Module, applying median row centering and Euclidian distance.

**Re-analysis of Xenopus E2F4 Chip-seq and RNA-seq.** RNA-seq (samples GSM1434783 to GSM1434788) and ChIP-seq (samples GSM1434789 to GSM1434792) data were downloaded from GSE59309. Reads from RNA-seq were aligned to the Xenopus laevis genome release 7.1 using TopHat2[47] with default parameters. Quantification of genes was then performed using HTSeq-count[48] release 0.6.1 with "-m intersection-nonempty" option. Normalization and statistical analysis were performed using Bioconductor package DESeq2[49]. Differential expression analysis was done between Multicilin-hGR alone versus Multicilin-hGR

in the presence of E2f4ΔCT. Reads from ChIP-seq were mapped to the X. laevis genome release 7.1 using Bowtie2[50]. Peaks were called and annotated according to their positions on known exons with HOMER[51]. Peak enrichments of E2F4 binding site in the promoters of centriole genes and cell cycle genes[9] were estimated in presence or absence of Multicilin and a ratio of E2F4 binding (Multicilin vs no Multicilin) was calculated.

**Promoter reporter studies.** The human CDC20B promoter was cloned into the pGL3 Firefly Luciferase reporter vector (Promega) with SacI and NheI cloning sites. The promoter sequenced ranged from −1073 to +104 relative to the transcription start site. 37.5 ng of pGL3 plasmid were applied per well. pCMV6-Neg, pCMV6-E2F1 (NM_005225) and pCMV6-E2F4 (NM_001950) constructs were from Origene. 37.5 ng of each plasmid was applied per well. 25 ng per well of pRL-CMV (Promega) was applied in the transfection mix for transfection normalization (Renilla luciferase). HEK 293T cells were seeded at 20,000 cells per well on 96-well plates. The following day, cells were transfected with the indicated plasmids (100 ng of total DNA) with lipofectamine 3000 (Invitrogen). After 24 h, cells were processed with the DualGlo kit (Promega) and luciferase activity was recorded on a plate reader.

**Proximity ligation assays.** Fully differentiated HAECs were dissociated by incubation with 0.1% protease type XIV from S. griseus (Sigma-Aldrich) in HBSS (Hanks' balanced salts) for 4 h at 4 °C. Cells were gently detached from Transwells® by pipetting and then transferred to a microtube. Cells were then cytocentrifuged at 72×g for 8 min onto SuperFrostPlus slides using a Shandon Cytospin 3 cytocentrifuge. Slides were fixed for 10 min in methanol at −20 °C for Centrin2 and ZO1 assays, and for 10 min in 4% paraformaldehyde at room temperature and then permeabilized with 0.5% Triton X-100 in PBS for 10 min for acetylated-α-tubulin assays. Cells were blocked with 3% BSA in PBS for 30 min. The incubation with primary antibodies was carried out at room temperature for 2 h. Then, mouse and rabbit secondary antibodies from the Duolink® Red kit (Sigma-Aldrich) were applied and slides were processed according to manufacturer's instructions. Images were acquired using the Olympus Fv10i confocal imaging systems with ×60 oil immersion objective and Alexa 647 detection parameters.

**Animals.** All experiments were performed following the Directive 2010/63/EU of the European parliament and of the council of 22 September 2010 on the protection of animals used for scientific purposes. Experiments on X. laevis and mouse were approved by the 'Direction départementale de la Protection des Populations, Pôle Alimentation, Santé Animale, Environnement, des Bouches du Rhône' (agreement number F 13 055 21). Mouse experiments were approved by the French ethical committee no.14 (permission number: 62-12112012). Timed pregnant CD1 mice were used (Charles Rivers, Lyon, France).

**Immunostaining on mouse ependyma.** Dissected brains were subjected to 12 min fixation in 4% paraformaldehyde, 0.1% Triton X-100, blocked 1 h in PBS, 3% BSA, incubated overnight with primary antibodies diluted in PBS, 3% BSA, and incubated 1 h with secondary antibodies at room temperature. Ependyma were dissected further and mounted with Mowiol before imaging using an SP8 confocal microscope (Leica microsystems) equipped with a ×63 oil objective. The same protocol was used to prepare samples for super-resolution acquisition. Pictures were acquired with a TCS SP8 STED ×3 microscope equipped with an HC PL APO 93×/1.30 GLYC motCORR™ objective (Leica microsystems). Pericentrin was revealed using Alexa 514 (detection 535–564 nm, depletion 660 nm), γ-tubulin was revealed using Alexa 568 (detection 582–667 nm, depletion 775), and FOP was revealed using Alexa 488 (detection 498–531 nm, depletion 592 nm). Pictures were deconvoluted using Huygens software. Maximum intensity projection of 3 deconvoluted pictures is presented in Fig. 4g. Primary antibodies: rabbit anti-CDC20B (1:500; Proteintech, 133376-1-AP), mouse IgG anti-PLK1 (1:500; ThermoFisher, 33–1700), rabbit anti-Pericentrin (1:500, Abcam, ab4448), mouse IgG1 anti-FoxJ1 (1:1000; eBioscience, 14–9965), rabbit anti-Deup1 (1:1000; kindly provided by Dr Xueliang Zhu), rabbit anti-Deup1 (1:250; Proteintech, 24579-1-AP), mIgG1 anti-γ-Tubulin (clone GTU88) (1:250; Abcam, Ab 11316), rabbit anti-ZO1 (1:600; ThermoFisher, 61–7300), rabbit anti-Spag5 (1:500; Proteintech, 14726-1-AP), mouse IgG1 anti-ZO1 (1:600; Invitrogen, 33-9100), mouse IgG2b anti-FGFR1OP (FOP) (1:2000; Abnova, H00011116-M01), mouse IgG1 anti-α-tubulin (1:500; Sigma-Aldrich, T9026). Secondary antibodies: Alexa Fluor 488 goat anti-rabbit (1:800; ThermoFisher Scientific, A-11034), Alexa Fluor 647 goat anti-rabbit (1:800; ThermoFisher Scientific, A-21244), Alexa Fluor 514 goat anti-rabbit (1:800; ThermoFisher Scientific, A-31558), Alexa Fluor 488 goat anti-mouse IgG2b (1:800; ThermoFisher Scientific, A-21141), Alexa Fluor 568 goat anti-mouse IgG2b (1:800; ThermoFisher Scientific, A-21144), Alexa Fluor 488 goat anti-mouse IgG2a (1:800; ThermoFisher Scientific, A-21131), Alexa Fluor 568 goat anti-mouse IgG1 (1:800; ThermoFisher Scientific, A-21134), Alexa Fluor 647 goat anti-mouse IgG1 (1:800; ThermoFisher Scientific, A-21240).

**Mouse constructs.** Expression constructs containing shRNA targeting specific sequences in the CDC20B coding sequence under the control of the U6 promoter

were obtained from Sigma-Aldrich (ref. TRCN0000088273 (sh273), TRCN0000088274 (sh274), TRCN0000088277 (sh277)). PCX-mcs2-GFP vector (Control GFP) kindly provided by Xavier Morin (ENS, Paris, France), and U6 vector containing a validated shRNA targeting a specific sequence in the NeuroD1 coding sequence[52] (Control sh, ref. TRCN0000081777, Sigma-Aldrich) were used as controls for electroporation experiments.

**Post-natal mouse brain electroporation.** The detailed protocol for post-natal mouse brain electroporation established by Boutin and colleagues[53] was used with minor modifications. Briefly, P1 pups were anesthetized by hypothermia. A glass micropipette was inserted into the lateral ventricle, and 2 μL of plasmid solution (concentration 3 μg/μL) was injected by expiratory pressure using an aspirator tube assembly (Drummond). Successfully injected animals were subjected to five 95 V electrical pulses (50 ms, separated by 950 ms intervals) using the CUY21 edit device (Nepagene, Chiba, Japan), and 10 mm tweezer electrodes (CUY650P10, Nepagene) coated with conductive gel (Signagel, Parker laboratories). Electroporated animals were reanimated in a 37 °C incubator before returning to the mother.

**Statistical analyses of mouse experiments.** Analysis of CDC20B signal intensity in deuterosomes (dot plot in Fig. 3b). For each category, >25 cells from two different animals were analyzed. Deuterosome regions were delineated based on FOP staining and the intensity of CDC20B fluorescent immunostaining was recorded using ImageJ software, and expressed as arbitrary units. Unpaired $t$ test vs immature: $p = 0.0005$ (intermediate, ***); $p < 0.0001$ (Mature, ****).

Analysis of $Cdc20b$ shRNAs efficiency (Fig. 4c): For each cell at the deuterosomal stage, the intensity of CDC20B fluorescent immunostaining was recorded using ImageJ software and expressed as arbitrary units. Data are mean ± sem. Two independent experiments were analyzed. A minimum of 35 cells per condition was analyzed. $n = 3, 4, 5$ and 5 animals for sh control, sh273, sh274, and sh277, respectively. Unpaired $t$ test vs sh control: $p < 0.0001$ (sh273, sh274, and sh277 ****).

Analysis of the number of FOXJ1-positive cells at 5dpe (Fig. 4d): Unpaired $t$ test vs sh control: 0.3961 (sh273, ns), 0.1265 (sh274, ns), 0.3250 (sh277, ns).

Analysis of the number of cells with non-disengaged centrioles at 9dpe (Fig. 4g): 15–20 fields were analyzed per condition. $n = 4, 4, 3$, and 4 animals for sh control, sh273, sh274, and sh277, respectively, from two independent experiments. Unpaired $t$ test vs sh control: $p < 0.0001$ (sh273, sh274, and sh277 ****).

Analysis of the number of centrioles per cell at 15dpe (Fig. 4j): > 100 cells were analyzed per condition. $n = 3, 3, 3$, and 3 animals for sh control, sh273, sh274, and sh277, respectively, from two independent experiments. Unpaired $t$ test vs sh control: $p < 0.0001$ (sh273, sh274, sh277 ****).

Analysis of ependymal cell categories at 15dpe (Fig. 4k): Data are mean ± sem from three independent experiments. More than 500 cells were analyzed for each condition. $n = 4, 4, 3$, and 3 animals for sh control, sh273, sh274, and sh277, respectively. Unpaired $t$ test vs sh control: $p = 0.0004$ (sh273, ***), 0.0001 (sh274, ****), 0.0038 (sh277, **).

**Mouse tracheal epithelial cells (MTECs).** MTECs cell cultures were established from the tracheas of 12 weeks-old mice. After dissection, tracheas were placed in cold DMEM:F-12 medium (1:1) supplemented with 15 mM HEPES, 100 U/mL penicillin, 100 μg/mL streptomycin, 50 μg/mL gentamicin sulfate, and 2.5 μg/mL amphotericin B. Each trachea was processed under a binocular microscope to remove as much conjunctive tissue as possible with small forceps and was opened longitudinally with small dissecting scissors. Tracheas were then placed in supplemented DMEM:F-12 containing 0.15% protease XIV from *S. griseus*. After overnight incubation at 4 °C, FCS was added to a final concentration of 10%, and tracheal epithelial cells were detached by gentle agitation. Cells were centrifuged at 400 g for 10 min and resuspended in supplemented DMEM:F-12 containing 10% FCS. Cells were plated on regular cell culture plates and maintained in a humidified atmosphere of 5% $CO_2$ at 37 °C for 4 h to allow attachment of putative contaminating fibroblast. Medium containing cells in suspension was further centrifuged at $400 \times g$ for 5 min and cells were resuspended in supplemented DMEM:F-12 containing BEGM Singlequots kit supplements and 5% FCS. Cells were plated on rat tail collagen I-coated Transwell®. Typically, 5 tracheas resulted in 12 Transwells®. Medium was changed every other day. Air–liquid interface culture was conducted once transepithelial electrical resistance had reached a minimum of 1000 ohm/cm² (measured with EVOM2, World Precision Instruments).

Air–liquid interface culture was obtained by removing medium at the apical side of the Transwell®, and by replacing medium at the basal side with supplemented DMEM:F-12 containing 2% Ultroser-G$^{TM}$ (Pall Corporation). 10 μM DAPT (N-[N-(3,5-difluorophenacetyl)-L-alanyl]-S-phenylglycine t-butyl ester) (Sigma) was added one day after setting-up the air–liquid interface.

**Immunostaining on HAECs and MTECs.** Three days after setting-up the air–liquid interface, MTECs on Transwell membranes were pre-extracted with 0.5% Triton X-100 in PBS for 3 min, and then fixed with 4% paraformaldehyde in PBS for 15 min at room temperature. HAECs were treated 21 days after setting-up the air–liquid interface. They were fixed directly on Transwells® with 100% cold methanol for 10 min at −20 °C (for CDC20B and Centrin2 co-staining,

Supplementary Figure 4a, b) or with 4% paraformaldehyde in PBS for 15 min at room temperature (for CDC20B single staining, Supplementary Figure 4c). All cells were then permeabilized with 0.5% Triton X-100 in PBS for 5 min and blocked with 3% BSA in PBS for 30 min. The incubation with primary and secondary antibodies was carried out at room temperature for 2 h and 1 h, respectively. Nuclei were stained with 4,6-diamidino-2-phenylindole (DAPI). Transwell® membranes were cut with a razor blade and mounted with ProLong Gold medium (Thermo-Fisher). Primary antibodies: rabbit anti-CDC20B (1:500; Proteintech, 133376-1-AP), rabbit anti-DEUP1 (1:500; Proteintech, 24579-1-AP), anti-Centrin2 (Clone 20H5, 1:500; Millipore, 04-1624). Secondary antibodies: Alexa Fluor 488 goat anti-rabbit (1:1000; ThermoFisher Scientific, A-11034), Alexa Fluor 647 goat anti-mouse (1:1000; ThermoFisher Scientific, A-21235). For co-staining of CDC20B and DEUP1, CDC20B primary antibody was directly coupled to CF$^{TM}$ 633 with the Mix-n-Stain$^{TM}$ kit (Sigma-Aldrich) according to the manufacturer's instruction. Coupled primary antibody was applied after secondary antibodies had been extensively washed and after a 30 min blocking stage in 3% normal rabbit serum in PBS.

**Western blot and immunofluorescence on transfected cells.** Cos-1 or Hela cells cells were grown in DMEM supplemented with 10% heat inactivated FCS and transfected with Fugene HD (Roche Applied Science) according to manufacturer's protocol. Transfected or control cells were washed in PBS and lysed in 50 mM Tris HCl pH 7.5, 150 mM NaCl, 1 mM EDTA, containing 1% NP-40 and 0.25% sodium deoxycholate (modified RIPA) plus a Complete Protease Inhibitor Cocktail (Roche Applied Science) on ice. Cell extracts separated on polyacrylamide gels were transfered onto Optitran membrane (Whatman) followed by incubation with rabbit anti-mouse CDC20B (1:500, Proteintech, 24579-1-AP) or homemade rabbit anti-*Xenopus* Cdc20b (1:300) antibody and horseradish peroxidase conjugated secondary antibody (Jackson Immunoresearch Laboratories, 711-035-152 and 715-035-150). Signal obtained from enhanced chemiluminescence (Western Lightning ECL Pro, Perkin Elmer) was detected with MyECL Imager (ThermoFisher Scientific).

For immunofluorescence staining, transfected cells were grown on glass coverslips and fixed for 6 min in methanol at −20 °C. Cells were washed in PBS, blocked in PBS, 3% BSA and stained with rabbit anti-*Xenopus* Cdc20b (1:300) or rabbit anti-CFTR (1:200, Santa-Cruz Biotechnology, 10747) as a negative control, in blocking buffer. After washings in PBS 0.1% Tween-20, cells were incubated with Alexa fluor 488 donkey anti-rabbit antibody (ThermoFisher Scientific, R37118), washed, and DNA was stained with 250 ng/mL DAPI. Coverslip were then rinsed and mounted in Prolong Gold antifade reagent (ThermoFisher Scientific) and confocal images were acquired by capturing Z-series with 0.3 μm step size on a Zeiss LSM 510 laser scanning confocal microscope.

**Co-immunoprecipitation studies.** Asynchronous HEK cells transfected with the plasmids described below, using lipofectamine 3000 according to manufacturer's instructions, were rinsed on ice with chilled Ca2+ and Mg2+ free Dulbecco's PBS (DPBS, Invitrogen), harvested using a cell scraper and lysed on ice for 5 min in lysis buffer (0.025 M Tris, 0.15 M NaCl, 0.001 M EDTA, 1% NP-40, 5% glycerol; pH 7.4) supplemented with EDTA and Halt™ Protease and Phosphatase Inhibitor Cocktail (Pierce, ThermoFisher). Lysates were clarified (12,000×$g$, 4 °C, 10 min) and the protein concentrations were determined using the Bradford assay (Bio-Rad). Immunoprecipitations were performed with the Pierce co-immunoprecipitation kit (Pierce, ThermoFisher) according to the manufacturer's instructions. For each immunoprecipitation, 1–1.5 mg of total lysate was precleared on a control column, then incubated on columns coupled with 20 μg of anti-GFP or anti-c-myc antibody (clone 9E10). Incubation was performed overnight at 4 °C. Columns were washed and eluted with 50 μL elution buffer. Samples were denatured at 70 °C for 10 min with Bolt™ LDS Sample Buffer and Bolt reducing agent, then separated on 4–12% gradient Bolt precast gels (ThermoFisher), transferred onto nitrocellulose (Millipore), and subjected to immunoblot analysis using either anti-CDC20B (ProteinTech, 133376-1-AP, 1/500) or anti-c-myc antibody (clone 9E10, 1/1000). In Fig. 6, note that the high level of expression of myc-PLK1 (Fig. 6a) and myc-SPAG5 (Fig. 6b) drained out locally the ECL reagent at the peak of the protein. The resulting double bands correspond to unique ones. Human SPAG5, sub-cloned into pCMV6-MT, was from OriGene. Human DEUP1 and PLK1 were cloned into pCS2-MT vector (Addgene). Human CDC20B was cloned into pEGFP-C1, pEGFP-N1 (Clontech) for the GFP fusion protein and pIRES-EYFP (Addgene) for the untagged protein.

**In-gel digestion, NanoHPLC, and Q-exactive plus analysis.** For mass spectrometry analysis, protein spots were manually excised from the gel and destained with 100 μL of H2O/ACN (1/1). After 10 min vortexing, liquid was discarded, and the procedure was repeated 2 times. They were rinsed with acetonitrile and dried under vacuum. Extracts were reduced with 50 μL of 10 mM dithiothreitol for 30 min at 56 °C, then alkylated with 15 μL of 55 mM iodoacetamide for 15 min at room temperature in the dark. They were washed successively by: (i) 100 μL of H2O/ACN (1/1) (2 times) and (ii) 100 μL of acetonitrile. Gel pieces were rehydrated in 60 μL of 50 mM NH4HCO3 containing 10 ng/μL of trypsin (modified porcine trypsin, sequence grade, Promega) incubated for one hour at 4 °C. After the

removal of trypsin, samples were incubated overnight at 37 °C. Tryptic peptides were extracted with: (i) 60 μL of 1% FA (formic acid) in water (10 min at RT), (ii) 60 μL acetonitrile (10 min at RT). Extracts were pooled, concentrated under vacuum, resuspended in 15 μL of aqueous 0.1% formic acid for NanoHPLC separation.

Separation was carried out using a nanoHPLC (Ultimate 3000, ThermoFisher Scientific). After concentration on a μ-Precolumn Cartridge Acclaim PepMap 100 $C_{18}$ (i.d. 5 mm, 5 μm, 100 Å, ThermoFisher Scientific) at a flow rate of 10 μL/min, using a solution of $H_2O$/ACN/FA 98%/2%/0.1%, a second peptide separation was performed on a 75 μm i.d. × 250 mm (3 μm, 100 Å) Acclaim PepMap 100 $C_{18}$ column (ThermoFisher Scientific) at a flow rate of 300 nL/min. Solvent systems were: (A) 100% water, 0.1% FA, (B) 100% acetonitrile, 0.08% FA. The following gradient was used $t = 0$ min 6% B; $t = 3$ min 6% B; $t = 119$ min, 45% B; $t = 120$ min, 90% B; $t = 130$ min 90% B (temperature at 35 °C).

NanoHPLC was coupled via a nanoelectrospray ionization source to the Hybrid Quadrupole-Orbitrap High Resolution Mass Spectrometer (ThermoFisher Scientific). MS spectra were acquired at a resolution of 70,000 (200 $m/z$) in a mass range of 300–2000 $m/z$ with an AGC target 3e6 value of and a maximum injection time of 100 ms. The 10 most intense precursor ions were selected and isolated with a window of 2 $m/z$ and fragmented by HCD (Higher energy C-Trap Dissociation) with normalized collision energy (NCE) of 27. MS/MS spectra were acquired in the ion trap with an AGC target 2e5 value, the resolution was set at 17 500 at 200 $m/z$ combined with an injection time of 100 ms.

Data were reprocessed using Proteome Discoverer 2.1 equipped with Sequest HT. Files were searched against the Swissprot Homo sapiens FASTA database (update of February 2016). A mass accuracy of ±10 ppm was used to precursor ions and 0.02 Da for product ions. Enzyme specificity was fixed to trypsin, allowing at most two miscleavages. Because of the previous chemical modifications, carbamidomethylation of cysteines was set as a fixed modification and only oxydation of methionine was considered as a dynamic modification. Reverse decoy databases were included for all searches to estimate false discovery rates, and filtered using the Percolator algorithm at a 1% FDR.

**Xenopus embryo injections, plasmids, RNAs, and morpholinos.** Eggs obtained from NASCO females were fertilized in vitro, dejellied and cultured using standard protocols[54]. All injections were done at the 8-cell stage in one animal-ventral blastomere (presumptive epidermis), except for electron microscopy analysis for which both sides of the embryo were injected, and for RT-PCR analysis for which 2-cell embryos were injected.

cdc20b riboprobe was generated from X. laevis cDNA. Full-length sequence was subcloned in pGEM™-T Easy Vector Systems (Promega). For sense probe, it was linearized by SpeI and transcribed by T7. For antisense probe it was linearized by ApaI and transcribed by Sp6 RNA polymerase. Synthetic capped mRNAs were produced with the Ambion mMESSAGE mMACHINE Kit. pCS105/GFP-CAAX was linearized with AseI and mRNA was synthesized with Sp6 polymerase. pCS2-mRFP and pCS2-GFP-gpi were linearized with NotI and mRNA was synthesized with Sp6 polymerase. pCS-Centrin4-YFP (a gift from Reinhard Köster, Technische Universität Braunschweig, Germany) was linearized with NotI and mRNA was synthesized with Sp6 polymerase. pCS2-GFP-Deup1 and pCS2-Multicilin(MCI)-hGR were kindly provided by Chris Kintner; both plasmids were linearized with ApaI, and mRNAs were synthesized with Sp6 polymerase. Embryos injected with MCI-hGR mRNA were cultured in Dexamethasone 20 μM in MBS 0,1× from st11 until fixation. pCS2-Separase wild-type and phosphomutant 2/4 (protease dead, PD) were provided by Marc Kirchner and Olaf Stemann, respectively; plasmids were linearized with NotI and mRNAs were synthesized with Sp6 polymerase. Venus-cdc20b, cdc20b-Venus, and cdc20b were generated by GATEWAY™ Cloning Technology (GIBCO BRL) from Xenopus laevis cdc20b cDNA. cdc20b was also subcloned in pCS2-RFP to make RFP-cdc20b and cdc20b-RFP fusions. All cdc20b constructs were linearized with NotI and mRNAs were synthesized with Sp6 polymerase. Quantities of mRNA injected: 500 pg for GFP-CAAX, RFP, GFP-gpi, Separase and Separase(PD); 25 to 500 pg for GFP-Deup1; 40 to 500 pg for MCI-hGR; 1 ng for Venus-cdc20b, cdc20b-Venus, cdc20b, and cdc20b-RFP; 500 pg to 1 ng for RFP-cdc20b.

Two independent morpholino antisense oligonucleotides were designed against cdc20b (GeneTools, LLC). cdc20b ATG Mo: 5′-aaatcttctctaacttccagtccat-3′, cdc20b Spl Mo 5′-acacatggcacaacgtacccacatc-3′. 20 ng of MOs was injected per blastomere or 10 ng of each Mo for co-injection.

**PCR and quantitative RT-qPCR.** Xenopus embryos were snap frozen at different stages and stored at −80 °C. Total RNAs were purified with a Qiagen RNeasy kit (Qiagen). Primers were designed using Primer-BLAST Software. PCR reactions were carried out using GoTaq® G2 Flexi DNA Polymerase (Promega). RT reactions were carried out using iScript™ Reverse Transcription Supermix for RT-qPCR (BIO-RAD). qPCR reactions were carried out using SYBRGreen on a CFX Bio-rad qPCR cycler. To check cdc20b temporal expression by qPCR we directed primers to exons 9/10 junction (Forward: 5′-ggctatgaattggtgcccg-3′) and exons 10/11 junction (Reverse: 5′-gcagggagcagatctggg-3′) to avoid amplification from genomic DNA. The relative expression of cdc20b was normalized to the expression of the housekeeping gene ornithine decarboxylase (ODC) for which primers were as follows: forward: 5′-gccattgtgaagactctctccattc-3′: reverse: 5′-ttcgggtgattccttgccac-3′.

To check the efficiency of Mo SPL, expected to cause retention of intron 1 in the mature mRNA of cdc20b we directed forward (5′-cctcccgagagttagagga-3′) and reverse (5′-gcatgttgtacttttctgctcca-3′) primers in exon 1 and exon2, respectively.

To check the expression of p53 in morphants by qPCR, primers were as follows: forward: 5′-cgcagccgctatgagatgatt-3′; reverse: 5′-cacttgcggcacttaatggt-3′. The relative expression of p53 was normalized to Histone4 expression (H4) for which primers were as follows: forward: 5′-ggtgatgccctggatgttgt-3′; reverse: 5′-ggcaaaggaggaaaaggactg-3′.

**Immunostainining on Xenopus embryos.** Embryos were fixed in 4% paraformaldehyde (PFA) overnight at 4 °C and stored in 100% methanol at -20 °C. Embryos were rehydrated in PBT and washed in MABX (Maleic Acid Buffer + Triton X100 0,1% v/v). Next, embryos were incubated in Blocking reagent (Roche) 2% BR + 15% Serum + MABX with respective primary and secondary antibodies. The anti-Xenopus laevis CDC20B antibody was obtained by rabbit immunization with the peptide SPDQRRIFSAAANGT (amino acids 495–509) conjugated to keyhole limpet hemocyanin, followed by affinity purification (Eurogentec). For immunofluorescence, embryos were fixed at RT in PFA 4% in PBS, and incubated in the CDC20B antibody diluted 1/150 in BSA 3% in PBS. For all experiments, secondary antibodies conjugated with Alexa were used. GFP-CAAX in Supplementary Figure 5g was revealed using a rabbit anti-GFP antibody together with a secondary antibody coupled to Alkaline Phosphatase (AP), which was revealed as follows: embryos incubated with the AP-conjugated antibody were washed twice in alkaline phosphatase buffer (PAB) (NaCl 0.1 M, Tris HCl pH 9.5 0.1 M, $MgCl_2$ 0.05 M, Tween 0.1%), 10 min each. Next, embryos were incubated in PAB with INT/BCIP substrate (Roche, REF:11681460001) until appropriate staining. Finally embryos were washed twice in MABX and fixed in MEMFA 30 min at RT. To mark cortical actin in MCCs, embryos were fixed in 4% paraformaldehyde (PFA) in PBT (PBS + 0.1% Tween v/v) for 1 h at room temperature (RT), washed 3 × 10 min in PBT at RT, then stained with phalloidin-Alexa Fluor 555 (Invitrogen, 1:40 in PBT) for 4 h at RT, and washed 3 × 10 min in PBT at RT. Primary antibodies: mouse anti-Acetylated−α-Tubulin (Clone 6-11B-1, Sigma-Aldrich, T7451, 1:1000), rabbit anti-γ-Tubulin (Abcam, Ab 16504, 1:500), mouse anti-γ-Tubulin (Clone GTU88, Ab 11316, Abcam, 1:500), Chicken anti-GFP (AVES, GFP-1020, 1:1000), rabbit anti-GFP (Torrey Pines Biolabs, TP401, 1:500), mouse anti-Centrin (Clone 20H5, EMD Millipore, 04-1624, 1:500). Secondary antibodies: donkey anti-rabbit-AP (Jackson ImmunoResearch, 711055152, 1:1000), Alexa Fluor 647 goat anti-mouse IgG2a (1:500; ThermoFisher Scientific, A-21241), Alexa Fluor 488 goat anti-chicken (1:500; ThermoFisher Scientific, A-11039), Alexa Fluor 568 goat anti-rabbit (1:500; ThermoFisher Scientific, A-11011).

**In situ hybridization on Xenopus embryos.** Whole-mount chromogenic in situ hybridization and whole-mount fluorescent in situ hybridization (FISH) was performed as detailed by Marchal and colleagues[54], and Castillo-Briceno and Kodjabachian[55], respectively. For single staining, all RNA probes were labeled with digoxigenin. For FISH on section, embryos were fixed in 4% paraformaldehyde (PFA), stored in methanol for at least 4 h at −20 °C, then rehydrated in PBT (PBS + Tween 0.1% v/v), treated with triethanolamine and acetic anhydride, incubated in increasing sucrose concentrations and finally embedded with OCT (VWR Chemicals). 12 μm-thick cryosections were made. Double FISH on sections was an adaptation of the whole-mount FISH method. 80 ng of cdc20b digoxigenin-labeled sense and antisense riboprobes and 40 ng of antisense α-tubulin fluorescein-labeled riboprobe[56] were used for hybridization. All probes were generated from linearized plasmids using RNA-labeling mix (Roche). FISH was carried out using Tyramide Signal Amplification – TSA TM Plus Cyanine 3/Fluorescein System (Perkin Elmer). Antibodies: Anti-DigAP (Roche, 11266026, 1:5000), Anti-DigPOD (Roche, 11207733910, 1:500), Anti-FluoPOD (Roche, 11426346910, 1:500).

**Microscopy.** Confocal: Flat-mounted epidermal explants were examined with a Zeiss LSM 780 confocal microscope. Four-colors confocal z-series images were acquired using sequential laser excitation, converted into single plane projection and analyzed using ImageJ software. Scanning Electron Microscopy (SEM): stage 37 Xenopus embryos were fixed in 3% glutaraldehyde in 0.1 M phosphate buffer pH 7.4 (19 mL monosodium phosphate 0.2 M and 81 mL disodium phosphate 0.2 M) made with filtered (0.22 μm) bi-distilled water, during 4 h with vigorous agitation, then washed with phosphatase buffer and filtered bi-distilled water, to be successively dehydrated in ethanol at 25, 50, and 70% for 30 min each; then, embryos were stored in fresh ethanol 70% at 4 °C for 1–2 days before further processing. Embryos in 70% ethanol were further dehydrated with vigorous agitation in ethanol once at 90% and twice at 100% for 30 min each; they were subsequently subjected to $CO_2$ critical point drying (CPD030, Balzers) at 31 °C and 73 atm. Finally, samples were sputter-coated with gold (vacuum 1 × 10–12 Torr, beam energy 3–4 keV) for immediate SEM digital imaging (FEI TENEO) of the skin epidermis. Transmission Electron Microscopy (TEM): stage 25 Xenopus embryos were fixed overnight at 4 °C in 2.5% glutaraldehyde, 2% paraformaldehyde, 0.1% tannic acid in a sodium cacodylate buffer 0.05 M pH 7.3. Next, embryos were washed 3 × 15 min in cacodylate 0.05 M at 4 °C. Post-fixation was done in 1% osmium buffer for 2 h. Next, embryos were washed in buffer for 15 min. Then, embryos were washed in water and dehydrated conventionally with alcohol, followed by a step in 70% alcohol containing 2% uranyl during 1 to 2 h at RT, or

overnight at 4 °C. Following three incubations in 100% alcohol, completed with three washes of acetone, embryos were included in classical epon resin, which was polymerized in oven at 60 °C for 48 h. Sections of 80 nm were made and analyzed into an FMI TECNAI microscope with acceleration of 200 kV.

**Statistical analysis of *Xenopus* experiments**. To quantify the effect of our different experiments, we applied one-way ANOVA analysis and Bonferroni's multiple comparisons test ($t$ test). ***$p < 0.05$; ns = not significant. Statistical analyses were done using GraphPad Prism 6.

Figure 5o and Fig. S6k: 10 cells per condition were analyzed and the total number of Centrin-YFP or γ-tubulin-positive spots per injected cell was counted.

Figure 7g: 5 fields (×20 zoom) per condition were analyzed, and the total number of properly ciliated MCCs based on acetylated α-tubulin staining among GFP positive cells per field was counted. Each field corresponded to a different embryo.

Figure 5s: 160–200 cells per condition were analyzed. $n = 6$, 8, and 10 embryos from three independent experiments for control, Mo ATG and Mo Spl, respectively. Unpaired $t$ test vs control: $p = 0.0037$ (Mo ATG **) and 0.0004 (Mo Spl ***).

## Data availability

scRNA-seq data were submitted to the GEO portal under series reference GSE103518. Proteomics data are available via ProteomeXchange with identifier PXD010629. All other relevant data are available from the authors.

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

## Acknowledgements

We are grateful to Chris Kintner, Marc Kirschner, Olaf Stemmann, Reinhard Köster, Xavier Morin and Xueliang Zhu for reagents. Imaging in IBDM was performed on PiCSL-FBI core facility, supported by the French National Research Agency through the program "Investments for the Future" (France-BioImaging, ANR-10-INBS-04). Sequencing at UCAGenomiX (IPMC), a partner of the National Infrastructure France Génomique, was supported by Commissariat aux Grands Investissements (ANR-10-INBS-09-03, ANR-10-INBS-09-02) and Canceropôle PACA. The authors thank Florian Roguet for Xenopus care, and Nathalie Garin from Leica Microsystems GmbH for technical advice on STED microscopy. We are grateful to Rainer Waldmann, Kévin Lebrigand, Virginie Magnone and Nicolas Nottet for fruitful discussions on single cell RNA sequencing, and Delphine Debayle for help with mass spectrometry experiments. We thank Julien Royet and Harold Cremer for insightful comments on the manuscript. This project was funded by grants from ANR (ANR-11-BSV2-021-02, ANR-13-BSV4-0013, ANR-15-CE13-0003), FRM (DEQ20141231765, DEQ20130326464, DEQ20180339158), Fondation ARC (PJA 20161204865, PJA 20161204542), the labex Signalife (ANR-11-LABX-0028-01), the association Vaincre la Mucoviscidose (RF20140501158, RF20120600738, RF20150501288), and the Chan Zuckerberg Initiative (Silicon Valley Fundation, 2017-175159-5022). OM, CB, and DRR were supported by fellowships from Ligue Nationale contre le Cancer (OM and CB), and Fondation ARC (DRR).

## Author contributions

P.B., B.M., and L.K. designed and supervised the study, and obtained funding. L.E.Z., S.R.G., and O.M. performed and analyzed human and mouse airway cells experiments. D.R.R. and V.T. performed and analyzed Xenopus experiments. C.B. performed and analyzed all experiments on mouse ependymal MCCs and contributed to the description of Xenopus deuterosomes. O.R. characterized CDC20B antibodies. M.D. and A.P. performed the bioinformatic analysis. N.P. carried out scRNA-seq experiments. A.S.G. performed mass spectrometry analyses. G.P. designed and performed CDC20B interaction studies. All authors were involved in data interpretation. D.R.R., L.E.Z., and C.B. designed the figures. L.K. drafted the original manuscript. D.R.R., L.E.Z., C.B., B.M., P.B., and L.K. edited the manuscript.

## Additional information

**Competing interests:** The authors declare no competing interests.

