## [Peer Review File · Nature Communications]

Reviewers' comments:

Reviewer #1 (Remarks to the Author):

Revinski et al describes the role of CCDC20B in centriole production in vertebrate MCCs. They report that CDC20B is found in a genome locus with other MCC regulators and using single cell RNA seq they find that its expression co-segregates with other genes involved in MCC differentiation/function. They propose that CDC20B protein localizes to the MCC specific deuterosome where it is involved in regulating centriole separation from the deuterosome. They present a very nice functional analysis in both mouse ependymal cells and xenopus epidermal that convincingly shows that CCDC20B has an important role in the proper development of MCCs. The link with separase is quite interesting however, perhaps a bit underdeveloped. Overall, I think this is an interesting paper that represents an important advance in our understanding of deuterosomes and MCC formation, however, there are some issues that concern me that I think should be resolved prior to publication. First and foremost the paper does a poor job of distinguishing between deuterosomes and centrioles. This is critical as they claim that CDC20B is involved in the separation of these two structures. The localization data is unconvincing and there is no actual analysis of centriole separation or lack thereof. Furthermore, many of the quantifications appear qualitative, which can be fine if it is properly defined what is being analyzed. "Normal" versus "not Normal" needs to be scientifically defined.

Specific Comments:

1. I find the localization data in Figure 2 a bit confusing. First, I am not sure what the authors are considering co-localization. In 2A Deup1 localizes to both large foci and small foci, but CDC20B seems to primarily be at the large foci. Are the small foci centrioles? Are they smaller deuterosomes that don't have CDC20B? This should be addressed with co-staining of centrioles together with deuterosomes. In 2B the authors use FOP as a marker for centrioles which it seems co-localizes with CDC20B whereas Deup1 created a smaller circle so would not have co-localized with CDC20B. Finally, the data from Xenopus appears diffuse compared to Deup1, a whole cell image with a centriole marker to distinguish between deuterosomes and centrioles would allow them to make the proper claims regarding CDC20B deuterosome localization. The Duolink assay is interesting, but suggests centriolar and cilia localization rather than deuterosome localization. It is unclear what is the potential role for CDC20B in the cilia, and if this underlies some of the ciliogenesis phenotypes. While a detailed ciliary analysis of CDC20B is beyond the scope of this manuscript I think that it should be mentioned in discussing the observed ciliogenesis phenotypes. If the authors want to make the claim that CCDC20B is at the deuterosome they need to have more convincing data.
2. Figure 3e-h. It not clear to me how deuterosome defects were quantified. How were deuterosomes distinguished from centrioles? What constitutes a deuterosome for the analysis (FOP? Which is a centriole marker?), what is considered normal size variation, what is considered minute or oversized. Similarly, it is unclear what constitutes a normal versus an abnormal MCC in 3K.
3. Figure 4a-c. I am a little confused by what is presented here. I understand that there is acetylated tubulin present but it is not making cilia, however I am confused by the GFP channel. All the cells look very unhealthy and I am not even sure which cells are the MCCs. Please explain or find more representative images.
4. I am confused by the discrepancy between centriole quantification (average ~150 centrioles/cell) in Figure 3o and gamma tubulin quantification (average ~275/cell) in Figure S4k. I have seen gamma tubulin used as a centriole marker but these numbers don't mesh, so what else is gamma tubulin staining? If it is inaccurate as a centriole marker then what is the analysis in S4k supposed to show us?
5. The authors claim that "Late-stage morphant MCCs that lacked cilia often displayed stalled deuterosomal figures, again suggesting a blockage in the centriole biosynthesis pathway". The figure shows gamma tubulin staining. Is gamma tubulin known to be at deuterosomes? How do they distinguish between stalled deuterosomes and undocked centrioles?

6. The super resolution of FOP and PCNT in 5G needs better explanation or labeling. I am not sure that I agree with their interpretation of PCNTs localization relative to FOP. In all their confocal images FOP is a ring, which makes it a poor marker for individual nascent centrioles. The super res might be better but it still does not seem to be identifying individual centrioles very clearly. To say that PCNT is specifically associated with centriolar walls seems like a stretch.

7. Figure 5n it is unclear what constitutes a "normal" MCC.

8. I like the model that CDC20B is involved in separase activity. The functional rescue of "normal" MCCs in cells deficient of CDC20B but overexpressing separase is striking and very interesting. However, I am very concerned regarding their interpretation of their data given that they never actually look at deuterosomes and centrioles together to determine if there is in fact a separation issue. This represents a gaping hole in the paper.

Reviewer #2 (Remarks to the Author):

In this manuscript, Revinski and coworkers performed single cell sequencing and identified a human airway epithelial cell population at the stage of centriole amplification. They found an enrichment of cell cycle-related gene transcripts in this population and investigated potential functions of CDC20B in centriole release from the deuterosome because its paralog CDC20 is known to activate APC/C and thus lead to Separase-dependent centriole disengagement and chromosome disjunction in mitotic cells. They found that Cdc20B was associated with deuterosomes during centriole biogenesis, in addition to localizations at mature basal bodies and in motile cilia of multiciliated cells (MCCs). Depletion of Cdc20B by RNAi in mouse ependyma repressed basal body production and multiciliogenesis and appeared to cause abnormal deuterosomes. In embryonic epidermis of *Xenopus* Cdc20B morphants, centrioles were clustered and failed to be polarized to the plasma membrane to serve as basal bodies. These defects were rescued upon overexpression of Separase in the morphants. As Cdc20B unlikely functions through the APC/C pathway based on its secondary structure analysis and literature survey, they propose that it functions in PLK1/Separase/PCNT pathway to facilitate centriole release.

The idea that Cdc20B could regulate centriole release echoes a recent publication in *Science*, in which the basal body production process in MCCs is found to be similarly regulated by proteins that control the centrosome cycle in cycling cells. Using mouse ependyma and frog embryonic epidermis as experimental systems, the authors obtained consistent results that suggest a role of Cdc20B in centriole release. The overall studies, however, suffer serious flaws, including misuse of markers in microscopy, poor data quality, inappropriate interpretation of results, and lack of a plausible mechanism. I thus do not recommend publication of this manuscript in *Nature Communications*, at least in its present form.

Detailed comments are listed below:

Major concerns:

Fig. 2: Where CDC20B is located is still confusing, due to inadequate markers and poor image resolution. The authors used MTECs, mouse ependymal cells, and *Xenopus* embryonic epidermis, but results in none of them is sufficiently informative. In MTECs and the epidermis (Fig. 2a, c) I do not agree with the claim that CDC20B associates with the deuterosome due to poor data quality. In the ependymal cells, there appears to be some relationship, though demonstrated indirectly using FOP, a mother centriole protein at the distal end, as marker. Even so, "association" is still a vague term. Important information such as (1) whether CDC20B localizes in the deuterosome or at certain region of the growing centrioles; (2) whether CDC20B exhibits dynamic associations with deuterosomes or centrioles are not addressed. These results will be important for understanding the claimed role of CDC20B in centriole release. Because the authors have access to STED microscope, I would suggest they costain Centrin, CDC20B, and Deup1 (or GFP-Deup1) and use STED microscopy to clarify these important questions.

Fig. 3: With less than 50% average knockdown efficiency, it is hard for me to be convinced of the deuterosome phenotypes. If the authors argue that Cdc20B was more thoroughly depleted in some cells than the others (Fig. 3b), they need to demonstrate the correlation between the phenotypes and Cdc20B depletion. Furthermore, FOP is not a deuterosome marker. They should have used Deup1 instead. The authors should also be cautious because, according to literature, deuterosome size and morphology varies a lot at different stages of centriole amplification. Fig. 3i-k indeed imply a role of Cdc20 in ciliogenesis. But again, FOP should not be used as deuterosome marker. Furthermore, although the authors used three different shRNAs, rescue experiment is always a better way to exclude off-target effects.

Fig. 4j-o. The Centrin signals are clustered together in the morphants. I doubt that the authors were able to distinguish single BBs at the shown image resolution in their quantification. Furthermore, if Cdc20B functions in centriole release, its depletion should not largely affect the centriole number.

Fig. S5h-j: The authors used r-tub foci in mitotic cells overexpressing Cdc20b or Separase to prove the proposed role of the axis in centriole disengagement. However, the data suffer serious problems. Firstly, centriole proteins such as centrin should have been used as marker because r-tub is located in PCM. In multipolar cells r-tub foci can each contain zero to several centrioles. Moreover, daughter centrioles prematurely released from their mothers in mitosis are unlikely capable of recruiting r-tub. Secondly, the overexpression may also impair chromosome cohesion and lead to abnormal mitosis. Therefore, the multiple r-tub foci, or centrosomes, may come from cells that failed to partition their centrosomes in the previous cell cycle.

Fig. S5k-w: A related question is why the epidermis still developed pretty well when the overexpression of Cdc20b or Separase from early embryonic stages led to serious mitotic defects as shown in S5h-j. I suggest that the authors briefly discuss this to reconcile their results.

Fig. 5: One really cannot tell whether the immunostaining signals in a-d were real or just background. How come the FOP signals are no longer ring-shaped in g?

Fig. 5o and discussion: The mechanism part is very weak. The authors hypothesized that Cdc20B functioned in the PLK1/Separase/PCNT axis instead of the APC/C/Serurin/Separase pathway. The strongest clue is that Cdc20B is reportedly associated with PLK1 in an interactome study. Others are just weak correlations. The authors consider the rescue experiments using Separase overexpression as a functional test for the hypothesis. Nevertheless, as the latter pathway also has a function in centriole release in MCCs, the results of the rescue experiments do not necessarily support their idea. Furthermore, even in their proposed scenarios that the phosphorylation of PCNT by PLK1 facilitates its degradation by Separase, the role of Cdc20B is still elusive: would the proposed Cdc20B-PLK1 interaction facilitate the activation of PLK1 or Separase or both? Therefore, additional data are definitely required to support their model.

Minor concerns:

1. Fig. S2: d and e should be swapped according to the text.
2. Data organization needs to be improved. For instance, Fig. S3k is described prior to Fig. S2f-k and even Fig. S3a-j. There are multiple such examples.
3. Page 4, line 17: Fig. S2b does not contain the described information.
4. Fig. S4e: the label "a-tub/Ac-tub" for the Y axis appears to represent "Ac-tub/a-tub".
5. Fig. S4f-k: r-tub is unlikely a good marker for BBs. It is a PCM protein and, to my knowledge, whether it is (solely) located on the BB in MCCs is not documented. Furthermore, its signals cannot be used to indicate deuterosome figures.
6. Fig. S5g: The labels for quantification method do not look right. For instance, "GFP/Actin Cap" probably means "Actin Cap/GFP".

Reviewer #3 (Remarks to the Author):

Review of CDC20b paper:

The manuscript by Revinski et al examines the role of Cdc20b in multiciliated cell (MCC) differentiation. The manuscript begins by presenting single cell RNAseq data in support of the general current view that many of components required during the cell cycle for centriole biogenesis, are also repurposed for MCC differentiation, since these cells undergo extensive centriole expansion. The manuscript then focuses on Cdc20b, an unannotated gene, which first came to light in their studies of miR449, as part of a locus containing several genes involved in MCC differentiation. The manuscript makes a very strong case that cdc20b is likely to be a conserved component of MCC differentiation, required when MCCs form in either the ependyma or *Xenopus* skin. This conclusion is well-supported by similar phenotypes shown in Figures 3 and 4, following knockdown experiments that were carried out in both ependymal cells and in the *Xenopus* skin, where in the latter case, the morpholino experiments are well controlled. These results make an important point that cdc20b, as well as Mir449a-c, in addition to the neighboring genes, mcdas and ccno, constitutes a conserved gene locus involved in MCC differentiation in different cell lineages.

To further support this conclusion, I would recommend better quantification of the basal body defect shown in Figure 3j, presenting the data as the number of basal bodies per cell as shown in Figure 4 for the *Xenopus* experiments, rather than using normal versus abnormal since the criteria for making this judgment are not clearly delineated.

However, the authors then try to explain their Cdc20b phenotype based on the model that Cdc20b, via plk1 and separase, is required for centriole release from the deuterosome. The data supporting this model are largely indirect and/or weak in their current form:

First, the authors claim that Cdc20b acts at the deuterosome but the data shown in Figure 2 supporting this claim are not very compelling. In panel A, there is very little overlap between the deup1 staining and the cdc20b staining. The large blob of deup1 staining is very unusual and not seen in other published deup1 staining patterns, making the overlap with Cdc20b as "deuterosome" localization suspect. The images in the rest of the panels are at such high power that it is not clear that the overlap reported is actually co-incident with the deuterosome, versus general cdc20b localization throughout the cytoplasm. In brief, these data are not sufficient to call cdc20b "a component of the vertebrate deuterosome" as stated in the Figure 2 Legend.

By contrast, the authors would seem to have stronger evidence that Cdc20b localizes strongly with basal bodies, and/or the axoneme based on the staining shown in Fig. S2. These data should be confirmed in *Xenopus* to provide confidence that this is localization is real, and if it is, should be highlighted in the main text, as it may be relevant to the phenotypes that they observe in the cdc20b knockdowns.

Second, in support of their model for Cdc20b action, the authors make claims about defects in deuterosome structures in their knockdown experiments. In ependymal cells, the effects of Cdc20b knock down on deuterosome formation reported in Figure 3f is based on FOP staining, but really needs to be done using Deup1 staining. In addition, the criteria to call the deuterosomes in knock-down cells "stalled", "oversized", "fused" and "minute" are not well explained and seem arbitrary since similar examples of such structures in the FOP staining are apparent in the sh control in Figure 3e. Given that the criteria used to call abnormal deuterosome staining is not clear, how this data is then quantified in Figure 3h is also not clear. The authors do not support their model of Cdc20b action in the *Xenopus* skin, where basal body production and/or docking seems to be

perturbed, but no evidence presented concerning deuterosome formation and/or basal body disengagement. Importantly, the author's model makes some strong predictions about the phenotype, where deuterosomes with procentrioles should form, elongate and mature, but then fail to disengage. However, there are no images presented showing that this indeed occurs in their phenotypes, either by super-resolution imaging or by TEM, in the ependyma or in the *Xenopus* skin.

Third, the author's model also suggests that *cdc20b* acts via *plk1* and *separase* for disengagement to occur. Again, the data presented to support this assertion are not as compelling as one would like. The staining in Figure 5a-c is not very informative since it is not carried out in a quantitative manner. As in Figure 2, PCNT or *Separase* appears not particularly localized to the deuterosomes based on the staining shown, but rather distributed throughout the cytoplasm. While an interaction between *Plk1* and *Cdc20b* might have been reported in a proteomics analysis, this interaction during MCC differentiation is not validated here. There are no mechanistic details as to how or whether *Cdc20b* affects *Plk1* activity during MCC differentiation. Perhaps the strongest support for the author's model comes from the *separase* rescue experiments shown in Figure 5h-n. However, these data shouldn't be quantified as normal MCCs, but rather by counting basal bodies per cell as done in Figure 4. The cilia staining is very spotty in the images, and if the authors were capturing images with 150 MCCs per field (Fig. 5n), then they would be unlikely to have a high enough resolution to quantify the phenotype in a meaningful way. Finally, even if *separase* were capable of rescuing the phenotype, it may have other activities during MCC differentiation other than disengagement. Since the authors have not shown that disengagement fails in their phenotype or that disengagement (versus some other aspect of basal body biogenesis) is then rescued by *separase* overexpression, the author's conclusions seem premature at this point.

In sum, I feel that the author's claims for a role for *Cdc20b* in MCC differentiation to be supported in the manuscript by the similar phenotypes that they see in ependymal cells and in the skin. However, the idea that *cdc20b* acts at the deuterosome to activate *Plk1* to allow centriole disengagement via *separase* is not supported by the data presented, making one hesitant to recommend publication in the current form.

Point-by-point response to reviewers

Reviewer #1 (Remarks to the Author):

Revinski et al describes the role of CCDC20B in centriole production in vertebrate MCCs. They report that CDC20B is found in a genome locus with other MCC regulators and using single cell RNA seq they find that its expression co-segregates with other genes involved in MCC differentiation/function. They propose that CDC20B protein localizes to the MCC specific deuterosome where it is involved in regulating centriole separation from the deuterosome. They present a very nice functional analysis in both mouse ependymal cells and xenopus epidermal that convincingly shows that CCDC20B has an important role in the proper development of MCCs. The link with separase is quite interesting however, perhaps a bit underdeveloped. Overall, I think this is an interesting paper that represents an important advance in our understanding of deuterosomes and MCC formation, however, there are some issues that concern me that I think should be resolved prior to publication. First and foremost the paper does a poor job of distinguishing between deuterosomes and centrioles. This is critical as they claim that CDC20B is involved in the separation of these two structures. The localization data is unconvincing and there is no actual analysis of centriole separation or lack there of. Furthermore, many of the quantifications appear qualitative, which can be fine if it is properly defined what is being analyzed. "Normal" versus "not Normal" needs to be scientifically defined.

Specific Comments:

I find the localization data in Figure 2 a bit confusing. First, I am not sure what the authors are considering co-localization. In 2A Deup1 localizes to both large foci and small foci, but CDC20B seems to primarily be at the large foci. Are the small foci centrioles? Are they smaller deuterosomes that don't have CDC20B? This should be addressed with co-staining of centrioles together with deuterosomes.

We agree that the image in the original Figure 2a did not clearly illustrate the association of CDC20B and DEUP1 staining. We have replaced it with a wider view of immature MCCs, and a zoom, where this association is much clearer (new Figure 3a).

Deuterosomes grow as they mature (Zhao et al., 2013). The preferential association of CDC20B to larger DEUP1 foci suggests that CDC20B enters the deuterosome environment at a late stage of the centriole amplification process. This point is now stressed in the revised text.

Related to our reviewer's concern, we have now included data showing reciprocal co-immunoprecipitation of CDC20B and DEUP1 upon co-transfection (new Figure 6c), supporting our interpretation of an association between CDC20B and deuterosomes.

In 2B the authors use FOP as a marker for centrioles which it seems co-localizes with CDC20B whereas Deup1 created a smaller circle so would not have co-localized with CDC20B.

We analyzed CDC20B relative to FOP, because the CDC20B and DEUP1 antibodies were both raised in rabbits, preventing double immunostaining. We had labeled CDC20B primary antibody to combine it with DEUP1 antibody, which proved feasible for mouse tracheal epithelial cells (MTECs), as shown in new Figure 3a. For this revision, we have attempted to apply this strategy to mouse ependymal cells (MEPCs). Unfortunately, the staining procedure proved too long to maintain the integrity of the explanted ependymal walls.

In MEPCs, active centriole synthesis around deuterosomes can easily be assessed through circular FOP staining (former Figure 2b), consistent with previous studies using Centrin2 to analyze deuterosomal figures (Al Jord et al., 2014 and 2017). Please note, however, that the FOP ring resolves into individual centrioles when resolution is improved with STED (new Figure 2b). In MEPCs, confocal imaging shows that CDC20B forms a ring nested in the FOP ring, and may only partially overlap with DEUP1. Thus, our reviewer is correct, and this is why we rather said that CDC20B is associated to the deuterosome. In the revised text, we have tried to clarify our interpretation by introducing the notion of perideuterosomal region. To further illustrate the relevance of CDC20B localization in the perideuterosomal region, we now show staining in this region of gamma-tubulin and Pericentrin, two key components of the pericentriolar material (new Figure 2a,b), which have been reported to localize close to Deup1 in BiOD assays (Firat-Karalar et al., 2017). This analysis reveals that centrioles growing around deuterosomes are embedded in perideuterosomal material, much like centrioles in the centrosome.

Finally, the data from *Xenopus* appears diffuse compared to Deup1, a whole cell image with a centriole marker to distinguish between deuterosomes and centrioles would allow them to make the proper claims regarding CDC20B deuterosome localization.

*We agree with our reviewer and strived to improve our demonstration in *Xenopus*. We now provide the first description of centriole amplification platforms in *Xenopus* (new Figure 2c,d). We show that these platforms contain g-tub, similar to*

the situation in MEPCs. Unlike in the mouse, however, deuterosomes in Xenopus epidermis do not appear as individual globular entities but more like large amorphous masses, in agreement with early electron microscopy studies (Steinman, 1969). Thus, our report sets the stage for future analysis of deuterosome-mediated centriole synthesis in Xenopus.

As requested, we now show whole cell images of RFP-Cdc20b/GFP-Deup1 co-expression together with Centrin or g-tub immunostaining (new Supp Figure 7h-m). We found that GFP-Deup1 globular structures could recruit Cdc20b, Centrin and g-tub, which confirms in Xenopus our newly reported interaction between CDC20B and DEUP1.

However, this approach was only partially satisfactory as it concerned exogenous Cdc20b. We thus decided to take advantage of a custom-made antibody against Xenopus Cdc20b. After validation (new Supp Figure 3b,c), this antibody was used in immunofluorescence to detect Cdc20b staining relative to GFP-Deup1 and Centrin in immature epidermal MCCs. We now show that Cdc20b accumulates in the vicinity of centriole amplification platforms, consistent with our analysis in MTECs and MEPCs (new Figure 3c).

The Duolink assay is interesting, but suggests centriolar and cilia localization rather than deuterosome localization. It is unclear what is the potential role for CDC20B in the cilia, and if this underlies some of the ciliogenesis phenotypes. While a detailed ciliary analysis of CDC20B is beyond the scope of this manuscript I think that it should be mentioned in discussing the observed ciliogenesis phenotypes. If the authors want to make the claim that CDC20B is at the deuterosome they need to have more convincing data.

As centrioles and cilia are more stable than deuterosomes, the detection in these places can be easier, notwithstanding a role of CDC20B to play at the deuterosomal stage, which is much more transient.

We could detect CDC20B near basal bodies and in cilia of human airway epithelial cells (HAECs)(new Supp Figure 4a-f) of MTECs and MEPCs (not shown). In contrast, our custom-made antibody detected Cdc20b only near basal bodies (new Supp Figure 4g-i) but not in cilia of Xenopus epidermal mature MCCs. Furthermore, neither N-terminal nor C-terminal Cdc20b fusions with GFP or RFP stained cilia in Xenopus. Since cilia are restored in Xenopus in Cdc20b-deficient MCCs after separate overexpression, Cdc20b is probably not required for ciliogenesis in this model, although it could be involved in proper cilia structure or function. Further work will be necessary to assess the impact of a CDC20B localization in cilia. As requested, we expanded the discussion about this specific point in the relevant

sections of the paper.

2. Figure 3e-h. It is not clear to me how deuterosome defects were quantified. How were deuterosomes distinguished from centrioles? What constitutes a deuterosome for the analysis (FOP? Which is a centriole marker?), what is considered normal size variation, what is considered minute or oversized.

Our intention was to score centriole amplification figures revealed by FOP staining, rather than deuterosomes themselves. However, the labeling of the graph in h was indeed misleading, and we apologize for this. We agree that this part of our analysis could have benefited from a more precise quantification. However, since it was not possible to relate aberrant morphologies of centriole amplification figures to specific outcomes in mature MCCs, we decided to remove those data from the paper. Instead, we have concentrated our efforts on a concern common to our three reviewers, which was the need to better document the disengagement problem that we had hypothesized. Thus, in place of former Figure 3e-h panels, we now show images and a graph that reveal the lack of disengagement of FOP-positive centrioles from DEUP1-positive deuterosomes in shCDC20B MCCs, at a stage when control cells have docked and aligned their centrioles and built cilia (new Figure 4e-g).

Similarly, it is unclear what constitutes a normal versus an abnormal MCC in 3K. *MCCs were scored abnormal when they did not display organized centriole patches associated to cilia. This is now clarified in the legend. As requested by another reviewer, we have now added a quantification of the number of centrioles per cell, which dropped on average by 50% in shCDC20B MCCs (new Figure 4j).*

3. Figure 4a-c. I am a little confused by what is presented here. I understand that there is acetylated tubulin present but it is not making cilia, however I am confused by the GFP channel. All the cells look very unhealthy and I am not even sure which cells are the MCCs. Please explain or find more representative images. *Membrane-bound GFP was used to identify injected cells. Control cells received only mGFP RNA, whereas morphant cells received mGFP RNA mixed with cdc20b morpholinos. The aspect of the GFP staining shown in this figure is not uncommon. Epidermal cells are loaded with secretory vesicles that can be stained by mGFP, which was particularly obvious in the morphant embryos shown here.*

4. I am confused by the discrepancy between centriole quantification (average ~150 centrioles/cell) in Figure 3o and gamma tubulin quantification (average

~275/cell) in Figure S4k. I have seen gamma tubulin used as a centriole marker but these numbers don't mesh, so what else is gamma tubulin staining? If it is inaccurate as a centriole marker then what is the analysis in S4k supposed to show us?

In Xenopus MCCs, gamma-Tubulin is located in two positions around the basal body revealed by Centrin (see figure below). The strongest spot likely corresponds to the basal foot, as it is localized posteriorly to the basal body, in the direction of ciliary stroke, and was associated to this structure in mouse tracheal MCCs (Clare et al., 2014). The second spot is weaker and localizes opposite to the strong spot. Z-projection reveals that this weaker spot is situated at the basis of the basal body. Resolving the precise localization of g-Tub will require immunogold staining, which was beyond the scope of this present study. In conclusion, each BB is associated to two spots of g-Tub. This is now clarified in the legend of new Supp Figure 6. Thus, the decrease in basal body numbers is confirmed by two different markers, which provided a stronger demonstration.

5. The authors claim that “Late-stage morphant MCCs that lacked cilia often displayed stalled deuterosomal figures, again suggesting a blockage in the centriole biosynthesis pathway”. The figure shows gamma tubulin staining. Is gamma tubulin known to be at deuterosomes? How do they distinguish between stalled deuterosomes and undocked centrioles?

We now show that g-tub is indeed found in centriole amplification platforms in

Xenopus (new Figure 2c,d). However, we agree that the distinction between stalled deuterosomes and undocked centrioles can be difficult to make. Similar to the analysis in MEPCs, we now provide evidence of Centrin-positive centrioles that failed to disengage from GFP-Deup1 positive platforms in Cdc20b-deficient MCCs, at a stage when normally all centrioles have docked, cilia are built and functional (new Figure 5p-u).

6. The super resolution of FOP and PCNT in 5G needs better explanation or labeling. I am not sure that I agree with their interpretation of PCNTs localization relative to FOP. In all their confocal images FOP is a ring, which makes it a poor marker for individual nascent centrioles. The super res might be better but it still does not seem to be identifying individual centrioles very clearly. To say that PCNT is specifically associated with centriolar walls seems like a stretch.

We do not concur with the conclusion that FOP is a poor marker of nascent centrioles. The main issue is indeed the insufficient level of resolution of confocal imaging. We have included in our revised version a photograph of FOP staining as revealed by STED, which provides a clear view of the circular organization of individual nascent centrioles (new Figure 2b). However, we agree that combining PCNT and FOP complicates interpretation. We have replaced our original picture by a new one, hoping that it will be easier to interpret for readers (new Figure 2b). We also modified the description in the text of PCNT distribution relative to FOP and now mention that PCNT stained fibers around growing procentrioles.

7. Figure 5n it is unclear what constitutes a “normal” MCC.

MCCs were scored as normal when multiple cilia are formed, as this revealed a correct number of centrioles being synthesized, docked and able to sustain axonemal growth.

8. I like the model that CDC20B is involved in separase activity. The functional rescue of “normal” MCCs in cells deficient of CDC20B but overexpressing separase is striking and very interesting. However, I am very concerned regarding their interpretation of their data given that they never actually look at deuterosomes and centrioles together to determine if there is in fact a separation issue. This represents a gaping hole in the paper.

We thank our reviewer for his/her appreciation of our model.

As explained above, we have added novel experimental evidence in both mouse and Xenopus that confirms the failure of many centrioles to disengage from DEUP1-positive deuterosomes in CDC20B knockdowned MCCs.

Reviewer #2 (Remarks to the Author):

In this manuscript, Revinski and coworkers performed single cell sequencing and identified a human airway epithelial cell population at the stage of centriole amplification. They found an enrichment of cell cycle-related gene transcripts in this population and investigated potential functions of CDC20B in centriole release from the deuterosome because its paralog CDC20 is known to activate APC/C and thus lead to Separase-dependent centriole disengagement and chromosome disjunction in mitotic cells. They found that Cdc20B was associated with deuterosomes during centriole biogenesis, in addition to localizations at mature basal bodies and in motile cilia of multiciliated cells (MCCs). Depletion of Cdc20B by RNAi in mouse ependyma repressed basal body production and multiciliogenesis and appeared to cause abnormal deuterosomes. In embryonic epidermis of *Xenopus* Cdc20B morphants, centrioles were clustered and failed to be polarized to the plasma membrane to serve as basal bodies. These defects were rescued upon overexpression of Separase in the morphants. As Cdc20B unlikely functions through the APC/C pathway based on its secondary structure analysis and literature survey, they propose that it functions in PLK1/Separase/PCNT pathway to facilitate centriole release.

The idea that Cdc20B could regulate centriole release echoes a recent publication in *Science*, in which the basal body production process in MCCs is found to be similarly regulated by proteins that control the centrosome cycle in cycling cells. Using mouse ependyma and frog embryonic epidermis as experimental systems, the authors obtained consistent results that suggest a role of Cdc20B in centriole release. The overall studies, however, suffer serious flaws, including misuse of markers in microscopy, poor data quality, inappropriate interpretation of results, and lack of a plausible mechanism. I thus do not recommend publication of this manuscript in *Nature Communications*, at least in its present form.

Detailed comments are listed below:

Major concerns:

Fig. 2: Where CDC20B is located is still confusing, due to inadequate markers and poor image resolution. The authors used MTECs, mouse ependymal cells, and *Xenopus* embryonic epidermis, but results in none of them is sufficiently

informative. In MTECs and the epidermis (Fig. 2a, c) I do not agree with the claim that CDC20B associates with the deuterosome due to poor data quality.

We have replaced the original set of pictures for MTECs by a new one that clearly shows the overlap between CDC20B and DEUP1 staining (new Figure 3a).

In this revised version, we have introduced data obtained with a custom-made antibody against Xenopus CDC20B. This antibody reveals the presence of Cdc20b close to centriole amplification platforms revealed by GFP-Deup1 (new Figure 3c).

In the ependymal cells, there appears to be some relationship, though demonstrated indirectly using FOP, a mother centriole protein at the distal end, as marker. Even so, "association" is still a vague term. Important information such as (1) whether CDC20B localizes in the deuterosome or at certain region of the growing centrioles; (2) whether CDC20B exhibits dynamic associations with deuterosomes or centrioles are not addressed. These results will be important for understanding the claimed role of CDC20B in centriole release. Because the authors have access to STED microscope, I would suggest they costain Centrin, CDC20B, and Deup1 (or GFP-Deup1) and use STED microscopy to clarify these important questions.

Early ultrastructure studies have revealed that deuterosomes consist of an inner dense core surrounded by fibrous material. Nascent centrioles grow within this perideuterosomal matrix. We now refer to these early descriptions when first introducing the deuterosome in the text. While Deup1 stains the inner core, we show in this revised version that the perideuterosomal matrix contains PCNT and γ -tubulin, two main components of the pericentriolar material (new Figure 2a,b). We actually also report with improved immunoreactivity that Deup1 also extends fibers in this matrix (new Figure 2a). To answer point (1) of our reviewer, CDC20B is primarily enriched in the perideuterosomal region rather than in the core. To answer point (2) of our reviewer, we have quantified the intensity of CDC20B signals in the region surrounding deuterosomes, and found that it becomes enriched there as deuterosomes mature (new Figure 3b). We thank our reviewer for this suggestion, as this result is consistent with the proposed role of CDC20B in centriole release.

Unfortunately, CDC20B and DEUP1 antibodies cannot be combined in MEPCs (see answer to Reviewer #1), for which we have implemented STED analysis. STED analysis was also used on HAECs but the mouse DEUP1 antibody shows poor immunoreactivity against human DEUP1. STED analysis has proven impossible so far on Xenopus epidermis, due to the presence of pigment.

Fig. 3: With less than 50% average knockdown efficiency, it is hard for me to be convinced of the deuterosome phenotypes. If the authors argue that Cdc20B was more thoroughly depleted in some cells than the others (Fig. 3b), they need to demonstrate the correlation between the phenotypes and Cdc20B depletion. *We understand our reviewer's concern and tried to address it. Thus, from the same set of images, we have quantified CDC20B signal intensity in individual MCCs at deuterosomal stages (new Figure 4c). This revealed a wide range of values for control cells, likely reflecting increased CDC20B levels with maturation time (new Figure 3b). Since MCCs do not differentiate in perfect synchrony in ependyma in vivo, that can create situations where shCDC20B cells displayed signals within the range of control cells. A clear point is that shCDC20B cells with the lowest signals did show obvious defects, and were selected in our report. We would like also to stress that three independent shRNAs gave similar results in terms of CDC20B depletion and phenotypes, which makes it very unlikely that the defects reported are due to off-targets. Thus, it appears that mouse ependymal MCC differentiation is quite sensitive to CDC20B depletion.*

Furthermore, FOP is not a deuterosome marker. They should have used Deup1 instead. The authors should also be cautious because, according to literature, deuterosome size and morphology varies a lot at different stages of centriole amplification.

We initially obtained a small amount of Deup1 antibody from X. Zhu, which prevented us from using it to analyse CDC20B depleted cells. For this revised version of the paper, we could avail of a very good commercial antibody, which allowed us to confirm that centrioles fail to disengage from deuterosomes in shCDC20B ependymal MCCs (new figure 4e-g).

Fig. 3i-k indeed imply a role of Cdc20 in ciliogenesis. But again, FOP should not be used as deuterosome marker. Furthermore, although the authors used three different shRNAs, rescue experiment is always a better way to exclude off-target effects.

CDC20B depletion in MEPCs severely impairs ciliogenesis. This could be secondary to defective basal body production or reflect a more direct function of CDC20B in ciliogenesis.

FOP was used in these panels to detect centrioles. We have now counted the number of individual centrioles in MCCs 15 days after electroporation, and found a decrease of 50% on average (new Figure 4j).

We are unaware of any experimental design to demonstrate a rescue against

shRNAs. Our major, and in our opinion strong, point is that the same phenotypes were observed with three independent shRNAs in mouse and after Cdc20b knockdown in Xenopus.

Fig. 4j-o. The Centrin signals are clustered together in the morphants. I doubt that the authors were able to distinguish single BBs at the shown image resolution in their quantification. Furthermore, if Cdc20B functions in centriole release, its depletion should not largely affect the centriole number.

The images shown are maximum intensity projections, but counting was performed manually on individual horizontal slices of 700nm.

Individual Centrin spots were counted, which were expected to represent individual centrioles. When centrioles remain clustered, the number of individual Centrin spots decrease. Thus, a drop in the number of Centrin spots remains consistent with a function of Cdc20b in centriole release.

Fig. S5h-j: The authors used r-tub foci in mitotic cells overexpressing Cdc20b or Separase to prove the proposed role of the axis in centriole disengagement. However, the data suffer serious problems. Firstly, centriole proteins such as centrin should have been used as marker because r-tub is located in PCM. In multipolar cells r-tub foci can each contain zero to several centrioles. Moreover, daughter centrioles prematurely released from their mothers in mitosis are unlikely capable of recruiting r-tub. Secondly, the overexpression may also impair chromosome cohesion and lead to abnormal mitosis. Therefore, the multiple r-tub foci, or centrosomes, may come from cells that failed to partition their centrosomes in the previous cell cycle.

We agree with our reviewer, and have reproduced this analysis with Centrin/g-tub/DAPI triple staining (see below). It appears that g-tub foci always contained two centrioles, which is compatible with failed mitosis in the previous cycle, rather than premature centriole disengagement. It remains that CDC20B and Separase induce the same defect, which supports their implication into a common functional axis. However, we decided to remove this data and the related text from the paper as it does not directly connect with the proposed mechanism.

Fig. S5k-w: A related question is why the epidermis still developed pretty well when the overexpression of Cdc20b or Separase from early embryonic stages led to serious mitotic defects as shown in S5h-j. I suggest that the authors briefly discuss this to reconcile their results.

The rate of abnormal mitosis observed in response to overexpression of Cdc20b or Separase remains fairly low, and does not reach a point where epidermis development is prevented. As mitotic defects are no longer presented in the paper, the revised text does not make reference to this explanation.

Fig. 5: One really cannot tell whether the immunostaining signals in a-d were real or just background. How come the FOP signals are no longer ring-shaped in g?

Those pictures revealed stronger signals in large cells (differentiating MCCs) than in small cells (undifferentiated progenitors), which we take as an indication that signals reflected expression, consistent with our scRNAseq data and published studies (Al Jord et al., 2017; Ma et al., 2014). In this revised version, we have decided to remove images of Securin and Separase immunostaining, as they did not strongly support enrichment in deuterosomes. Instead, we have added new pictures to show the association of PLK1 to deuterosomes, marked by DEUP1 (new Figure 6d).

STED imaging improved the resolution of nascent centrioles but FOP signal remains organized in a circle around deuterosomes.

Fig. 5o and discussion: The mechanism part is very weak. The authors hypothesized that Cdc20B functioned in the PLK1/Separase/PCNT axis instead of the APC/C/Serurin/Separase pathway. The strongest clue is that Cdc20B is reportedly associated with PLK1 in an interactome study. Others are just weak correlations. The authors consider the rescue experiments using Separase overexpression as a functional test for the hypothesis. Nevertheless, as the latter pathway also has a function in centriole release in MCCs, the results of the rescue experiments do not necessarily support their idea. Furthermore, even in their proposed scenarios that the phosphorylation of PCNT by PLK1 facilitates its degradation by Separase, the role of Cdc20B is still elusive: would the proposed Cdc20B-PLK1 interaction facilitate the activation of PLK1 or Separase or both? Therefore, additional data are definitely required to support their model.

We now provide experimental evidence through immunoprecipitation and mass spectrometry that CDC20B does not interact with APC/C components (new Table S2). Thus, CDC20B may not directly activate APC/C and Separase. We also show in this revised version through reciprocal co-immunoprecipitation that CDC20B does interact with DEUP1, PLK1 and SPAG5 (new Figure 6a-c). The interaction with DEUP1 is consistent with the presence of CDC20B around deuterosomes. Further supporting this view, we also show in Xenopus embryos that overexpressed RFP-CDC20B and GFP-Deup1 can be found associated in large globular centriole amplification entities (new Supp Figure 7h-m). The interaction of CDC20B with PLK1 initially detected in an unbiased interactome study is now experimentally confirmed in our manuscript. Likewise, the interaction of CDC20B with SPAG5 detected in another unbiased interactome study is also confirmed. SPAG5 was found associated to DEUP1 in a BioID assay (Firat-Karalar et al., 2014) and is a known regulator of Separase (Thein et al., 2007; Chiu et al., 2014). We now show that SPAG5 is detected in mature deuterosomes in MEPCs (new Figure 6e). This new set of evidence supports the view that CDC20B, SPAG5 and PLK1 indeed collaborate in mature deuterosomes where they are all detected. Based on previous knowledge, we proposed that centriole release from deuterosomes may mobilize Separase, and we indeed proved that Separase overexpression rescued centriole release in cells where Cdc20b was knockdowned. As PCNT is known to be targeted by Separase following phosphorylation by PLK1, we find particularly relevant the presence of PCNT in the perideuterosomal region. Our view is consistent with the recent report by Al Jord and colleagues (2017), which lends a role for PLK1 and APC/C in centriole disengagement. As requested by our reviewer, we now provide additional data that all concur to strengthen our initial model. It

remains unknown how exactly PLK1 and Separase are activated in deuterosomes, but the finding that CDC20B may bridge DEUP1, SPAG5 and PLK1 in these structures represents in our mind an important step forward in the field of multiciliogenesis.

Minor concerns:

1. Fig. S2: d and e should be swapped according to the text.

Done.

2. Data organization needs to be improved. For instance, Fig. S3k is described prior to Fig. S2f-k and even Fig. S3a-j. There are multiple such examples.

Following our reviewer's request, we paid a peculiar attention to improve organization, and we hope that the new one will be found clearer by our reviewer.

3. Page 4, line 17: Fig. S2b does not contain the described information.

Reference was made to the WB showing immunoreactivity of the human CDC20B antibody against the mouse protein.

4. Fig. S4e: the label "a-tub/Ac-tub" for the Y axis appears to represent "Ac-tub/a-tub".

Correction made.

5. Fig. S4f-k: r-tub is unlikely a good marker for BBs. It is a PCM protein and, to my knowledge, whether it is (solely) located on the BB in MCCs is not documented. Furthermore, its signals cannot be used to indicate deuterosome figures.

See response to reviewer #1 for the description of g-Tub staining at BBs.

We agree with our reviewer that it is unclear whether g-Tub, which we show to be linked to both deuterosome and basal bodies, could accurately reveal stalled deuterosomes; we have thus have removed the corresponding sentence. We now report stalled deuterosomal figures in Xenopus through combined GFP-Deup1 and Centrin staining (new figure 5p-u).

6. Fig. S5g: The labels for quantification method do not look right. For instance, "GFP/Actin Cap" probably means "Actin Cap/GFP".

Agreed and corrected as suggested.

Reviewer #3 (Remarks to the Author):

Review of CDC20b paper:

The manuscript by Revinski et al examines the role of Cdc20b in multiciliated cell (MCC) differentiation. The manuscript begins by presenting single cell RNAseq data in support of the general current view that many of components required during the cell cycle for centriole biogenesis, are also repurposed for MCC differentiation, since these cells undergo extensive centriole expansion. The manuscript then focuses on Cdc20b, an unannotated gene, which first came to light in their studies of miR449, as part of a locus containing several genes involved in MCC differentiation. The manuscript makes a very strong case that cdc20b is likely to be a conserved component of MCC differentiation, required when MCCs form in either the ependyma or Xenopus skin. This conclusion is well-supported by similar phenotypes shown in Figures 3 and 4, following knockdown experiments that were carried out in both ependymal cells and in the Xenopus skin, where in the latter case, the morpholino experiments are well controlled. These results make an important point that cdc20b, as well as Mir449a-c, in addition to the neighboring genes, mcidas and ccno, constitutes a conserved gene locus involved in MCC differentiation in different cell lineages.

To further support this conclusion, I would recommend better quantification of the basal body defect shown in Figure 3j, presenting the data as the number of basal bodies per cell as shown in Figure 4 for the Xenopus experiments, rather than using normal versus abnormal since the criteria for making this judgment are not clearly delineated.

We have followed this request and present quantification of individual centrioles/basal bodies in MEPCs. We found a decrease of 50% on average in shCDC20B MEPCs (new Figure 4j).

However, the authors then try to explain their Cdc20b phenotype based on the model that Cdc20b, via plk1 and separase, is required for centriole release from the deuterosome. The data supporting this model are largely indirect and/or weak in their current form:

First, the authors claim that Cdc20b acts at the deuterosome but the data shown in Figure 2 supporting this claim are not very compelling. In panel A, there is very little overlap between the deup1 staining and the cdc20b staining. The large blob of deup1 staining is very unusual and not seen in other published deup1 staining patterns, making the overlap with Cdc20b as “deuterosome” localization suspect.

This point was raised by all reviewers. Although the DEUP1 blob was reported by others (Zhao et al., 2013; Funk et al., 2015), it may not represent typical

deuterosomes. We have included new pictures in the revised version that show a clear overlap between CDC20B and DEUP1 staining (new Figure 3a). We also provide biochemical evidence of interaction between these two proteins (new Figure 6c).

The images in the rest of the panels are at such high power that it is not clear that the overlap reported is actually co-incident with the deuterosome, versus general cdc20b localization throughout the cytoplasm. In brief, these data are not sufficient to call cdc20b “a component of the vertebrate deuterosome” as stated in the Figure 2 Legend.

Please refer to our response to reviewers #1 and #2 regarding improved CDC20B localization data.

By contrast, the authors would seem to have stronger evidence that Cdc20b localizes strongly with basal bodies, and/or the axoneme based on the staining shown in Fig. S2. These data should be confirmed in *Xenopus* to provide confidence that this is localization is real, and if it is, should be highlighted in the main text, as it may be relevant to the phenotypes that they observe in the cdc20b knockdowns.

Please refer to our response to reviewer #1 regarding this point.

Second, in support of their model for Cdc20b action, the authors make claims about defects in deuterosome structures in their knockdown experiments. In ependymal cells, the effects of Cdc20B knock down on deuterosome formation reported in Figure 3f is based on FOP staining, but really needs to be done using Deup1 staining.

This point was common to all three reviewers and was experimentally addressed both in mouse and Xenopus with DEUP1 staining. Here again, we obtained very consistent data between our two models that clearly support our initial interpretation.

In addition, the criteria to call the deteuroosomes in knock-down cells “stalled”, “oversized”, “fused” and “minute” are not well explained and seem arbitrary since similar examples of such structures in the FOP staining are apparent in the sh control in Figure 3e. Given that the criteria used to call abnormal deteurosome staining is not clear, how this data is then quantified in Figure 3h is also not clear.

We agree that this analysis could seem arbitrary for the general reader, although the initial aim was to provide an unbiased view of the typical phenotypes recorded.

This part of our analysis was thus removed from the paper.

The authors do not support their model of Cdc20b action in the *Xenopus* skin, where basal body production and/or docking seems to be perturbed, but no evidence presented concerning deuterosome formation and/or basal body disengagement.

*We managed to titrate down GFP-Deup1 to reveal endogenous platforms of centriole amplification in *Xenopus* MCCs (see new Figure 2d). This allowed us to reveal the lack of disengagement of centrioles from deuterosomal platforms lacking Cdc20b (new Figure 5p-u).*

Importantly, the author's model makes some strong predictions about the phenotype, where deuterosomes with procentrioles should form, elongate and mature, but then fail to disengage. However, there are no images presented showing that this indeed occurs in their phenotypes, either by super-resolution imaging or by TEM, in the ependyma or in the *Xenopus* skin.

*We now provide such evidence in mouse and *Xenopus* using DEUP1 staining and confocal imaging. Super-resolution imaging generates rich but complicated images, and we did not feel that it would have further strengthen our interpretation. Since we are in the process of setting up SBF-SEM, future work will bring additional 3D information to be compared with TEM. Unfortunately, we did not manage to generate images of control and CDC20B knockdowned samples within the timeframe of this revision.*

Third, the author's model also suggests that cdc20b acts via plk1 and separase for disengagement to occur. Again, the data presented to support this assertion are not as compelling as one would like. The staining in Figure 5a-c is not very informative since it is not carried out in a quantitative manner.

The aim of this figure was to confirm the presence of PLK1, Securin and Separase in maturing MCCs. We now provide a picture at higher magnification to show the enrichment of PLK1 locally at the level of mature deuterosomes marked by DEUP1 (new Figure 6d).

As in Figure 2, PCNT or Separase appears not particularly localized to the deuterosomes based on the staining shown, but rather distributed throughout the cytoplasm.

This remark applies to Separase but not to PCNT that displays a clear enrichment in the region of deuterosomes. We now provide a view of PCNT on a single MCC at

deuterosome-stage in new Figure 2a that clearly makes this point. We further show in the same Figure that g-Tub also localizes in the perideuterosomal region. We also note that Separase is known to be distributed throughout the cytoplasm in interphasic cells. In fact, this type of localization in MCCs was reported for other relevant cell cycle proteins including CDK1, CyclinB1, and APC3 in cultured MEPCs (Al Jord et al., 2017).

While an interaction between Plk1 and Cdc20b might have been reported in a proteomics analysis, this interaction during MCC differentiation is not validated here.

We confirmed PLK1/CDC20B interaction through reciprocal co-immunoprecipitation in co-transfected cells (new Figure 6a). Confirming this interaction in native MCCs is a much more difficult task, which was beyond the scope of our present study. Regular co-immunoprecipitation requires lysis buffer which are knowledgeably not strong enough to access insoluble or high density cell structures. At present the best way to analyze endogenous interactions of proteins localized in insoluble cell structures like centrosomes (and probably deuterosomes) is proximity labeling assays such as BioID or APEX, which we plan to develop in the near future.

There are no mechanistic details as to how or whether Cdc20b affects Plk1 activity during MCC differentiation. Perhaps the strongest support for the author's model comes from the separase rescue experiments shown in Figure 5h-n. However, these data shouldn't be quantified as normal MCCs, but rather by counting basal bodies per cell as done in Figure 4. The cilia staining is very spotty in the images, and if the authors were capturing images with 150 MCCs per field (Fig. 5n), then they would be unlikely to have a high enough resolution to quantify the phenotype in a meaningful way. Finally, even if separase were capable of rescuing the phenotype, it may have other activities during MCC differentiation other than disengagement. Since the authors have not shown that disengagement fails in their phenotype or that disengagement (versus some other aspect of basal body biogenesis) is then rescued by separase overexpression, the author's conclusions seem premature at this point. ,

Cilia staining was evaluated at low magnification (see examples below) to record a large number of cells and selected high magnification pictures were shown in the figure for qualitative representation. In the revised manuscript, we show that centrioles disengagement fails in CDC20B knockdowned MCCs, both in mouse and Xenopus. In Xenopus, multiple cilia formation was recovered when Separase was

overexpressed in Cdc20b-deficient cells. As multiple cilia can only form from properly docked individual centrioles, we felt that reporting rescued multiciliogenesis both indicated that centriole disengagement was rescued but also that the entire differentiation process was recovered, which represented a more informative result.

In sum, I feel that the author's claims for a role for Cdc20b in MCC differentiation to be supported in the manuscript by the similar phenotypes that they see in ependymal cells and in the skin. However, the idea that cdc20b acts at the deuterosome to activate Plk1 to allow centriole disengagement via separase is not supported by the data presented, making one hesitant to recommend publication in the current form.

We hope that the new evidence provided in the revised manuscript will convince our reviewer that CDC20B is indeed involved in centriole disengagement in MCCs. Our work represents the first detailed study of this phenomenon across vertebrates. Although we could not demonstrate how CDC20B impacts on PLK1 activity, it suggests a plausible mechanistic scenario of centriole disengagement where PLK1 and Separase coordinate centriole disengagement, much like in cycling cells (Tsou and Stearns, 2009).

Reviewers' comments:

Reviewer #2 (Remarks to the Author):

The revised manuscript is largely improved in presentation. The authors have introduced a term perideuterosomal material (PDM) to describe proteins enriched in the corona region of deuterosomes and thus avoided the confusing in the initial manuscript. They have also removed inappropriate data and added new ones, including those showing clearly a centriole releasing defect upon Cdc20B knockdown (Fig. 4e-g; Fig. 5p-u) and associations among proposed players (Fig. 6), to strengthen their findings and make their model more plausible. Although the mechanism part is still weak, their major findings, such as a critical role of Cdc20B in centriole disengagement from deuterosomes and separate as a downstream target, are convincing and represent important advances. I thus recommend publication of the manuscript after minor revisions as detailed below.

Figure 2b: I cannot see where the "individual FOP-positive procentrioles" (line 116) or "growing procentrioles" (line 121) are from the STED images. If the authors can indeed define the positions of these procentrioles, please provide diagrams to help readers. Otherwise they should provide better images (costaining for centrin may help to locate the procentriole position; SIM may be better than STED for this purpose).

Figure 3: The manuscript says that in Fig. 3c "Cdc20b was found associated to Deup1-positive deuterosomes actively engaged in centriole synthesis" (line 149). These features, however, are not obvious to me. In their answers to reviewer #1's comments, the authors explain that deuterosomes in *Xenopus* epidermis appear more like large amorphous masses, in agreement with early electron microscopy studies (Steinman, 1969). Nevertheless, in Supplementary Fig. 7h-m, in which they examined exogenous Cdc20b, a typical deuterosome is clearly visible. I suggest that they reconcile these discrepancies. If the EM studies by Steinman support their results, they should cite the paper. In Fig. 3a, please indicate typical double-positive and DEUP1-alone foci with arrow/arrowhead.

Line 150-157: Supplementary Fig. 4 has little to do with the conclusion "CDC20B is tightly associated to mature deuterosomes". The descriptions should be removed to somewhere else for better readability.

I suggest to shift the data in Fig. 6d to Fig. 6b because they are described immediately following Fig. 6a in the main text.

Line 242: Fig. 6d should be 6e.

Reviewer #3 (Remarks to the Author):

The revised manuscript by Revinski et al addresses several concerns, by eliminating certain data while providing additional data to support the main conclusions. As stated in the original review, the phenotypes observed in various model systems certainly support the conclusion that Cdc20B is required for multiciliated cell differentiation. Whether or not Cdc20b acts as the authors propose in the manuscript is still an open question: the localization data for cdc20b is still not very solid, and the analysis of the phenotype less than compelling in terms of where the block in differentiation is occurring. Nonetheless, the current manuscript has a lot of data even if the proposed model for cdc20b action is not correct in every detail. Some minor outstanding concerns remain in the data presentation:

1) The lack of super-resolution imaging of cdc20b in the context of the deup1 as well other centriolar markers makes it hard to evaluate where it might be located. For example, the authors

refer to new "perideuterosomal" cdc20b localization. However, given that it is associated with the basal body, and its localization lies between the deup1 and the distal centriolar marker FOP, doesn't this place it more at the centriole than at the deuterosome?

2) The rationale for how the authors classified cdc20b staining as mature versus immature in Figure 3 is not entirely clear. As it is currently presented, it looks like the intensity of the FOP and Cdc20b staining might vary, but it is not clear how the authors decided what is mature and immature, implying that there is time or another structural element associated with maturity that was looked at as part of the analysis?

3) The overall conclusion is that the number of centrioles is reduced in the Cdc20b mutant because they are made but fail to be released. The quantification throughout the manuscript does not do a good job of supporting this conclusion. Is the number of deuterosomes, and associated centrioles formed in Cdc20b mutants the same as in wildtype? In Figure 4g, the actual percentage of cells affected per field is not presented, leaving one wondering whether it is a minor or significant phenotype. In Figure 4j, how does the number the centrioles released relate to the total number of centrioles present per cell? I can see that mature centriole number goes down, but it is very hard to determine whether this is primarily due to non-release, versus a reduction in deuterosome formation, and/or centriole nucleation.

4) TEM is used to analyze the phenotype in *Xenopus*, but focuses only on the undocked centrioles, rather than the centrioles that fail to release. Given the focus on the non-release phenotype in the manuscript, one would like to see what this looks like ultrastructurally both in *Xenopus* and in ependymal cells to see if it conforms to what the authors claim.

Response to Reviewers

Reviewer #2 (Remarks to the Author):

The revised manuscript is largely improved in presentation. The authors have introduced a term perideuterosomal material (PDM) to describe proteins enriched in the corona region of deuterosomes and thus avoided the confusing in the initial manuscript. They have also removed inappropriate data and added new ones, including those showing clearly a centriole releasing defect upon Cdc20B knockdown (Fig. 4e-g; Fig. 5p-u) and associations among proposed players (Fig. 6), to strengthen their findings and make their model more plausible. Although the mechanism part is still weak, their major findings, such as a critical role of Cdc20B in centriole disengagement from deuterosomes and separate as a downstream target, are convincing and represent important advances. I thus recommend publication of the manuscript after minor revisions as detailed below.

Figure 2b: I cannot see where the "individual FOP-positive procentrioles" (line 116) or "growing procentrioles" (line 121) are from the STED images. If the authors can indeed define the positions of these procentrioles, please provide diagrams to help readers. Otherwise they should provide better images (costaining for centrin may help to locate the procentriole position; SIM may be better than STED for this purpose).

To address this request, we have added a STED photograph taken from an ependymal MCC stained for FOP only, which may be easier to interpret. FOP staining accumulates along the walls of individual centrioles, which we have pointed out with arrowheads. 6 individual centrioles arranged in a circle could be unambiguously identified in this particular deuterosomal figure. We also provide a diagram carbon copied from this image, as suggested. Likewise, 6 individual centrioles could be identified and are now pointed out with arrowheads in the FOP/PCNT double stained deuterosomal figure.

Figure 3: The manuscript says that in Fig. 3c "Cdc20b was found associated to Deup1-positive deuterosomes actively engaged in centriole synthesis" (line 149). These features, however, are not obvious to me. In their answers to reviewer #1's comments, the authors explain that deuterosomes in *Xenopus* epidermis appear more like large amorphous masses, in agreement with early electron microscopy studies (Steinman, 1969). Nevertheless, in Supplementary Fig. 7h-m, in which they examined exogenous Cdc20b, a typical deuterosome is clearly visible. I suggest that they reconcile these discrepancies. If the EM studies by Steinman support their results, they should cite the paper. In Fig. 3a, please indicate typical double-positive and DEUP1-alone foci with arrow/arrowhead.

*Using g-tub/centrin double staining, we could molecularly analyze and report for the first time endogenous centriole amplification platforms in *Xenopus* epidermal MCCs. As shown in Fig. 2c, such platforms are heterogeneous in shape and size, consistent with Steinman's study, which was quoted in lines 58 and 126. When GFP-Deup1 was expressed at moderate levels (line 126) it formed numerous small granular structures within g-tub-positive masses (Fig. 2d). In contrast, when GFP-Deup1 was expressed at higher levels, it formed very few large and bright ring-shaped structures that did not resemble natural deuterosomes, as exemplified in Supp. Fig. 7 h-m (line 245: we carefully talked about "spherical structures*

positive for g-tub and centrin” and did not use the term deuterosome). This is consistent with the self-organization properties of Deup1 expressed in bacteria, as reported by Zhao and colleagues (2013) (quotation from their paper: “To further assess the importance of Deup1, we expressed polyhistidine (His)-tagged Deup1 in Escherichia coli (Fig. 7c). 3D-SIM indicated that His_Deup1 appeared as foci in the bacteria. At low protein levels, 81.7% of the foci (n=251) were granular, whereas at high levels 77.6% (n=183) formed ring-shaped structures (Fig. 7c,d).”).

Fig. 3a was labeled as requested.

Line 150-157: Supplementary Fig. 4 has little to do with the conclusion "CDC20B is tightly associated to mature deuterosomes". The descriptions should be removed to somewhere else for better readability.

The conclusion was meant for the entire section devoted to CDC20B protein distribution, and not only for Supp. Fig. 4 data, for which we do not see a better position in the paper. We have rephrased our sentence to avoid confusion.

I suggest to shift the data in Fig. 6d to Fig. 6b because they are described immediately following Fig. 6a in the main text.

We have followed this recommendation and have aligned PLK1 blots and immunofluorescence and have also done it for Spag5.

Line 242: Fig. 6d should be 6e.

Lettering in Fig. 6 has been changed, as suggested above by our reviewer, and has been carefully checked.

Reviewer #3 (Remarks to the Author):

The revised manuscript by Revinski et al addresses several concerns, by eliminating certain data while providing additional data to support the main conclusions. As stated in the original review, the phenotypes observed in various model systems certainly support the conclusion that Cdc20B is required for multiciliated cell differentiation. Whether or not Cdc20b acts as the authors propose in the manuscript is still an open question: the localization data for cdc20b is still not very solid, and the analysis of the phenotype less than compelling in terms of where the block in differentiation is occurring. Nonetheless, the current manuscript has a lot of data even if the proposed model for cdc20b action is not correct in every detail. Some minor outstanding concerns remain in the data presentation:

1) The lack of super-resolution imaging of cdc20b in the context of the deup1 as well other centriolar markers makes it hard to evaluate where it might be located. For example, the authors refer to new “perideuterosomal” cdc20b localization. However, given that it is associated with the basal body, and its localization lies between the deup1 and the distal centriolar marker FOP, doesn't this place it more at the centriole than at the deuterosome?

Following this comment, we have tried multiple times to use STED imaging on MTECs immunostained against Deup1, CDC20B (primary antibody coupled to fluorophore) and FOP (impossible to combine Deup1 and CDC20B immunostaining on ependymal MCCs). It proved impossible to retrieve images more informative than those already displayed in the revised manuscript.

We have defined the perideuterosomal region as the corona around the deuterosome core revealed by Deup1. Growing procentrioles are embedded in this corona, and CDC20B appears to be enriched in the proximal part of this corona. Thus, that CDC20B is present within the perideuterosomal region seems a safe assertion. Does the association of CDC20B to basal bodies of mature MCCs indicate that it is at the centrioles in amplification platforms? At first sight this argument could make sense but is in fact indirect and rather weak. As revealed by our pictures of mature MCCs in both mouse (Fig. 5e) and Xenopus (Fig. 5p), Deup1 is also associated to basal bodies. Yet, it is duly considered a specific deuterosome marker during the phase of centriole amplification.

It must also be stressed that virtually nothing was known about the precise architecture of deuterosome-based centriole amplification platforms and our paper provides new markers and the first STED images for some of the most robust and specific of those markers.

2) The rationale for how the authors classified cdc20b staining as mature versus immature in Figure 3 is not entirely clear. As it is currently presented, it looks like the intensity of the FOP and Cdc20b staining might vary, but it is not clear how the authors decided what is mature and immature, implying that there is time or another structural element associated with maturity that was looked at as part of the analysis?

Using FOP staining, we could indeed define a sequence of typical figures revealing MCC maturity, as shown below. At the “material accumulation stage” FOP staining was cloudy and very few deuterosomal figures could be discerned. Fig. 3b shows an MCC at this stage with a discernable deuterosomal figure devoid of CDC20B. Quite logically, our time-course analysis revealed that the frequency of material accumulation figures dramatically decreased between P3 and P15.

3) The overall conclusion is that the number of centrioles is reduced in the Cdc20b mutant because they are made but fail to be released. The quantification throughout the manuscript does not do a good job of supporting this conclusion. Is the number of deuterosomes, and associated centrioles formed in Cdc20b mutants the same as in wildtype? In Figure 4g, the actual percentage of cells affected per field is not presented, leaving one wondering whether it is a minor or significant phenotype. In Figure 4j, how does the number the centrioles released relate to the total number of centrioles present per cell? I can see that mature centriole number goes down, but it is very hard to determine whether this is

primarily due to non-release, versus a reduction in deuterosome formation, and/or centriole nucleation.

These are legitimate, yet very difficult questions to address. Is it possible to reliably count the number of deuterosomes? The answer is definitely no in Xenopus, as deuterosomes are heterogeneous in size and shape and could only be revealed with exogenous GFP-Deup1. It is likely not more reliable in mouse ependymal MCCs. MCCs do not mature synchronously in the ependymal tissue (photograph in 3b shows an immature MCC next to a fully mature ciliated MCC), and deuterosomes may not form synchronously in individual MCCs as different numbers can be counted in different cells (please compare Fig. 2a, 3b and 4f). Fig. 4f shows two typical cases of stalled centriole release of gradual severity. In the left cell, some but not all centrioles were released. In the right cell, up to 12 deuterosomes can be counted, which most probably indicates that deuterosomes formed correctly and nucleated centrioles but did not release them. This phenotype was quite penetrant, as revealed in graph 4g, now expressed as the percentage of MCCs with non-disengaged centrioles per field. Thus, it is impossible to rule out detrimental effects of CDC20B depletion on deuterosome assembly and centriole synthesis, but we provide good evidence that non-disengagement contributes significantly to the eventual phenotype. Moreover, in the current state of knowledge the rescue obtained through Separase overexpression also supports a role of CDC20B in centriole disengagement rather than in deuterosome or centriole synthesis.

4) TEM is used to analyze the phenotype in Xenopus, but focuses only on the undocked centrioles, rather than the centrioles that fail to release. Given the focus on the non-release phenotype in the manuscript, one would like to see what this looks like ultrastructurally both in Xenopus and in ependymal cells to see if it conforms to what the authors claim.

We agree that TEM would make a nice complement to our analysis of the non-release phenotype. It would not, however, change our conclusion that is based on statistically significant and consistent immunofluorescence results in two species. In contrast, TEM cannot be used as a statistically significant method of analysis.

Trying to follow our reviewer's suggestion, we went back to our original sections, and unfortunately did not find exploitable figures of non-disengaged centrioles. We would like to bring to the attention of our reviewer that MCCs compose just about 15% of the total number of outer layer epidermal cells, that not all MCCs may be injected with morpholinos and exhibit a non-disengaged phenotype, and that such figures may occupy a tiny fraction of the cytoplasm.

However frustrating it may be to us and our reviewer, we decided not to repeat the entire TEM analysis, as we feel that it would not significantly increase the impact of our study but would dramatically delay its publication.

REVIEWERS' COMMENTS:

Reviewer #2 (Remarks to the Author):

I am satisfied with the authors' revision and recommend publication of this manuscript.

Reviewer #3 (Remarks to the Author):

The authors might want to soften their conclusions about the primary function of Cdc20b. In mouse, the analyses thus far use conditions that are not full nulls: the knockdown of Cdc20B using siRNA in cultured cells is only partial (barely two-fold, Fig. 4c). Conversely, the phenotype in *Xenopus* may be closer to a null, but the potential off-target complications (even when these experiments are well controlled), and the amorphous nature of the deuterosome structures in the *Xenopus* cells along with the relatively low-resolution images shown in Fig. 5q,r,t, and u, leave open the question of whether the phenotype is actually due a defect in disengagement.

Given these uncertainties, I suggest that the authors should hedge their bets. The phenotype that arises in the mouse when Cdc20b is fully null may in the end emphasize different functions for Cdc20b than the one analyzed here. Furthermore, the disengagement defect reported here may be an indirect consequence of Cdc20b reduced function, since the authors have not ruled out defects in deuterosome formation/structure, or in centriole elongation/maturation, all of which may need to occur prior to, or in conjunction with, the process of disengagement. Finally, the authors should not discount the striking docking defect of disengaged centrioles and well as the strong localization of cdc20b to the basal body. The authors have done a good job at defining Cdc20b as a novel MMC differentiation factor, potentially used to regulate various steps in centriole biogenesis ,maturation and docking, with disengagement as just one possibility.

Response to reviewers

Reviewer #2 (Remarks to the Author):

I am satisfied with the authors' revision and recommend publication of this manuscript.

We thank our reviewer for his/her appreciation of our work.

Reviewer #3 (Remarks to the Author):

The authors might want to soften their conclusions about the primary function of Cdc20b. In mouse, the analyses thus far use conditions that are not full nulls: the knockdown of Cdc20B using siRNA in cultured cells is only partial (barely two-fold, Fig. 4c). Conversely, the phenotype in *Xenopus* may be closer to a null, but the potential off-target complications (even when these experiments are well controlled), and the amorphous nature of the deuterosome structures in the *Xenopus* cells along with the relatively low-resolution images shown in Fig. 5q,r,t, and u, leave open the question of whether the phenotype is actually due a defect in disengagement.

Knockdown in mouse was achieved in vivo through electroporation of anti-Cdc20b shRNAs in the postnatal brain, not in cultured cells. In Xenopus, potential off-target effects are unlikely to complicate interpretation as multiciliogenesis was efficiently rescued in cdc20b morphant embryos by cdc20b RNA injection.

Given these uncertainties, I suggest that the authors should hedge their bets. The phenotype that arises in the mouse when Cdc20b is fully null may in the end emphasize different functions for Cdc20b than the one analyzed here. Furthermore, the disengagement defect reported here may be an indirect consequence of Cdc20b reduced function, since the authors have not ruled out defects in deuterosome formation/structure, or in centriole elongation/maturation, all of which may need to occur prior to, or in conjunction with, the process of disengagement. Finally, the authors should not discount the striking docking defect of disengaged centrioles and well as the strong localization of cdc20b to the basal body. The authors have done a good job at defining Cdc20b as a novel MMC differentiation factor, potentially used to regulate various steps in centriole biogenesis, maturation and docking, with disengagement as just one possibility.

We agree with our reviewer that Cdc20b may carry out multiple functions during multiciliated cell differentiation. We had actually not discounted potential roles in disengaged centrioles and/or in cilia, and discussed that refined temporal inactivation would be required to address such roles. We also agree that alternative methods to suppress Cdc20b, such as a complete genetic knockout, may reveal earlier/additional functions than the one we document about centriole disengagement. We have significantly rewritten the first paragraph of our discussion to take these remarks into account.